# Replication stress induces POLQ-mediated structural variant formation throughout common fragile sites after entry into mitosis

Thomas E. Wilson [1,2] ✉, Samreen Ahmed[1,2], Amanda Winningham[2] & Thomas W. Glover [1,2] ✉

Genomic structural variants (SVs) greatly impact human health, but much is unknown about the mechanisms that generate the largest class of nonrecurrent alterations. Common fragile sites (CFSs) are unstable loci that provide a model for SV formation, especially large deletions, under replication stress. We study SV junction formation as it occurs in human cell lines by applying error-minimized capture sequencing to CFS DNA harvested after low-dose aphidicolin treatment. SV junctions form throughout CFS genes at a 5-fold higher rate after cells pass from G2 into M-phase. Neither SV formation nor CFS expression depend on mitotic DNA synthesis (MiDAS), an error-prone form of replication active at CFSs. Instead, analysis of tens of thousands of de novo SV junctions combined with DNA repair pathway inhibition reveal a primary role for DNA polymerase theta (POLQ)-mediated end-joining (TMEJ). We propose an important role for mitotic TMEJ in nonrecurrent SV formation genome wide.

Chromosomal rearrangements, i.e., structural variants (SVs), represent a large proportion of genomic diversity and are responsible for numerous genomic disorders[1]. They also arise in somatic cells and are a major mutation type in cancers. A predominant class of SVs are kb- to Mb-scale copy number variants (CNVs), including simple deletions and duplications and more complex rearrangements[2,3]. There are two major types of larger (>10 kb) CNVs in human genomes as revealed by breakpoint structures. Recurrent CNVs are formed by meiotic unequal recombination between flanking segmental duplications[4]. Non-recurrent germline CNVs and virtually all that arise in somatic and cancer cells can form anywhere in the genome and are characterized by short microhomologies or insertions at the breakpoint junctions[5,6].

Despite the large impact of nonrecurrent CNVs on human health, our understanding of the mechanisms responsible for their formation is incomplete. One challenge is that a single mechanism may not explain all SV events. Microhomologies at breakpoint junctions suggested early models of SV formation by nonhomologous end joining (NHEJ) of DNA double-strand breaks (DSBs)[7], since microhomologies cannot support the homologous recombination (HR) that drives recurrent CNV formation. However, some non-recurrent CNVs have multiple breakpoint junctions that are difficult to reconcile with NHEJ but might be explained by fork-stalling and template switching (FoSTeS), which invokes a DSB-independent switch of a nascent replication strand to a different template[8,9], or microhomology-mediated break-induced replication (MMBIR), which is similar to FoSTeS but invokes a single-ended DSB at a stalled replication fork[8,10]. Importantly, FoSTeS and MMBIR are thought to act during S-phase at sites of stalled replication whereas NHEJ predominates in G1[11].

Over many experiments, we established common fragile sites (CFSs) as an experimental model for non-recurrent CNV formation[12–18]. CFSs are genomic loci that show frequent gaps and breaks on metaphase chromosome spreads under replication stress. CFS expression is frequently induced by low-doses of the DNA B-family polymerase inhibitor aphidicolin (APH), but is also observed to arise spontaneously at a lower frequency[18,19]. CFSs were originally proposed to represent regions of DNA that remained unreplicated past S-phase[19] and decades of work support conclusions that CFSs correspond to the subset of late-replicating genomic loci residing at the largest human genes. When transcribed, these genes become susceptible to impaired

[1]Department of Pathology, University of Michigan Medical School, Ann Arbor, MI 48109, USA. [2]Department of Human Genetics, University of Michigan Medical School, Ann Arbor, MI 48109, USA. ✉e-mail: wilsonte@umich.edu; glover@umich.edu

replication fork progression due to a paucity of active replication origins resulting from transcription that continues into S-phase[17,20,21].

CFS loci are also hotspots for CNV formation under replication stress, both in human cancers[18] and in cell culture systems following treatment with multiple agents including low doses of APH, hydroxyurea, and ionizing radiation[15–17,22]. Like nonrecurrent CNVs in human genomes, CNVs formed at CFSs are characterized by short microhomologies, insertions, or blunt ends, leading us to propose they could be formed by template switching, NHEJ, or microhomology-mediated end joining[13]. An alternative potential mechanism was suggested by the findings of Minocherhomji et al[23]. who showed that APH-induced unreplicated genome regions complete replication in early mitotic prophase by a rescue process called mitotic DNA synthesis (MiDAS). Notably, peaks of MiDAS synthesis have a high correspondence with CFS genes[24,25]. Moreover, MiDAS often results in synthesis on just one sister chromatid suggesting it occurs by conservative break-induced replication (BIR)[26], a variant HR pathway operating at single-ended DSBs that is prone to template switching and other errors[27,28]. These observations link MiDAS to CFSs, possibly as the mechanism by which SV junctions form at these loci[29,30].

Most recently, a series of studies have described a specific form of end-joining catalyzed by DNA polymerase theta (POLQ), a pathway often called theta-mediated end-joining (TMEJ)[31]. TMEJ creates non-homologous junctions with a significant frequency of templated insertions that provide microhomologies for bridging two DSB ends. Intriguingly, POLQ activity and TMEJ become activated upon entry into mitosis through mechanisms that include RHINO-directed recruitment to mitotic DNA breaks and POLQ phosphorylation by the PLK1 kinase[32–35]. Thus, TMEJ joins NHEJ, FoSTeS, MMBIR, and MiDAS as a candidate mechanism for catalyzing SV junction formation at CFSs and elsewhere.

A limitation of prior literature is a paucity of prospective experimental tests of the hypothesized connections between replication stress, replication rescue and repair mechanisms, and SV formation based on direct assessment of de novo SV junctions. The high frequency of CNV formation at CFS loci provides a valuable experimental model for obtaining these important missing breakpoint junction data to add to indirect inferences from cancer genomes and cytological studies. We previously used whole genome microarrays to establish that large, actively transcribed CFS genes are hotspots for CNV formation, but throughput and resolution were low and CNVs could only be detected after clonal expansion[12–17]. More recently, we established svCapture sequencing as a reliable method for detecting and characterizing locus-specific single-molecule SV junctions[36].

Here, we used svCapture to enrich whole-genome sequencing within known CFS genes to determine the distribution, structure, breakpoint junctions, and mechanisms of SV formation during specific cell-cycle phases and in cells deficient in key repair processes. We reasoned that identifying the cell cycle phase at which SV junctions formed following APH-induced replication stress would guide the identification of the mechanism(s) involved since FoSTes and MMBIR would likely predominate in S-phase, MiDAS and TMEJ in M-phase, and NHEJ in the following G1 phase. We found that that many APH-induced SVs formed at CFSs during M-phase but that MiDAS was not required for this activity. In contrast, by examining SV frequencies in cell populations and the structure of tens of thousands of de novo CFS SV junctions, we identified TMEJ as a primary contributor to SV junction formation, with NHEJ having a lesser contribution. These results reveal a role for mitotic TMEJ in mediating large SV formation at CFSs and provide important insights into possible genome-wide SV mutation mechanisms in normal and cancer cells.

## Results

### svCapture detects large de novo deletion SVs in CFS genes

We used hybridization target capture, i.e., svCapture[36], to enrich whole-genome sequencing near the middle of five previously established CFS genes in three cell lines to enable detection of de novo single-molecule SV formation in cell populations (Fig. 1A). Specifically, we targeted large genes *PRKG1*, *NEGR1*, and *MAGI2* in fibroblast line UM-HF1 (HF1) and *FHIT* and *WWOX* in lymphoblastoid line GM12878 and colon cancer line HCT116 as model systems for replication stress-associated SV formation (Supplementary Fig. 1A and B). *PRKG1*, *NEGR1*, and *MAGI2* are known CNV hotspots in HF1 cells[17], whereas *FHIT* and *WWOX* are among the loci with the highest frequency of metaphase breaks and gaps in lymphoblastoid and HCT116 cells[37].

We first used a workflow in which SVs were allowed to accumulate asynchronously under replication stress caused by low doses of APH, an inhibitor of B-family replicative DNA polymerases α, δ and ε (Fig. 1B), determined empirically per cell line to slow but not stop replication and to produce an average of 2-5 chromosome gaps and breaks per cell. Throughout, we only scored SV junctions present in a single source DNA molecule to track de novo SVs formed during the experiment. Paired samples showed a clear increase in de novo SV Frequency, i.e., SV count normalized to target region coverage, in APH-treated vs. control cells (Fig. 1C). In some cell types, observed frequencies were consistent with as many as half of measured haplotypes acquiring a de novo SV (Fig. 1D), consistent with prior microarray results[17].

SVs induced at CFS genes were strongly biased toward deletions over duplications, inversions (Fig. 1D), and inter-target translocations, even though svCapture reports all SV junction types detectable by short-read sequencing. The lower level of non-deletion SVs was sometimes significantly induced by APH (Supplementary Fig. 1C–G), but due to the greater abundance of DNA loss events originating from unreplicated DNA[17,22], we mainly track deletions below. Figure 1E verifies reproducible deletion induction across multiple experimental batches (denoted by point color) in all cell lines (the single replicate of GM12878 cells is supported by further data below).

APH-induced deletion SVs showed mainly short microhomologies at breakpoint junctions and a median SV size of ~200 kb in all cell lines (Fig. 1C, F, and Supplementary Fig. 1C), matching the 186 kb median for microarray CNVs in human cell lines[13,17]. SVs tended to be smaller without APH induction and for duplications (Supplementary Fig. 1H and I). We therefore applied filters to only track SVs smaller than 1.2 Mb and larger than 10 kb (50 kb for inversions, Supplementary Fig. 1C) to maintain specificity.

### SVs formed under aphidicolin stress arise throughout large gene bodies

Proposed mechanisms for genomic instability at CFS genes variably invoke replication delay[17], local sequence features prone to polymerase stalling within CFS genes[38,39], and additional influences secondary to transcription[40,41]. Our high-density collection of SV breakpoints informs this longstanding question. As expected, we saw the greatest frequency of deletion breakpoints within capture target regions (Fig. 1G, Supplementary Fig. 2A and D). However, some of the second SV breakpoints were outside the capture targets but remained almost entirely within the gene bodies. Like prior work[17], duplications and inversions were more frequently located at the gene flanks as opposed to deletions, which clustered in the center of the genes (Supplementary Fig. 1D).

When we normalized deletion breakpoint density within capture targets to local sequencing coverage to account for variable svCapture efficiency and underlying clonal SVs (Fig. 1H, Supplementary Fig. 2B and E), we observed that breakpoints were distributed throughout the 250 kb or 400 kb capture targets (Fig. 1I, Supplementary Fig. 2C and F). No specific locations in any of the five targeted genes appeared to act as unexpectedly high frequency sites of focal SV formation. The pattern was consistent with replication forks failing stochastically throughout large, transcribed genes.

Together, svCapture recapitulated all aspects of SV formation at CFS genes under replication stress as previously seen in microarray data but with much greater resolution and data density. We do not

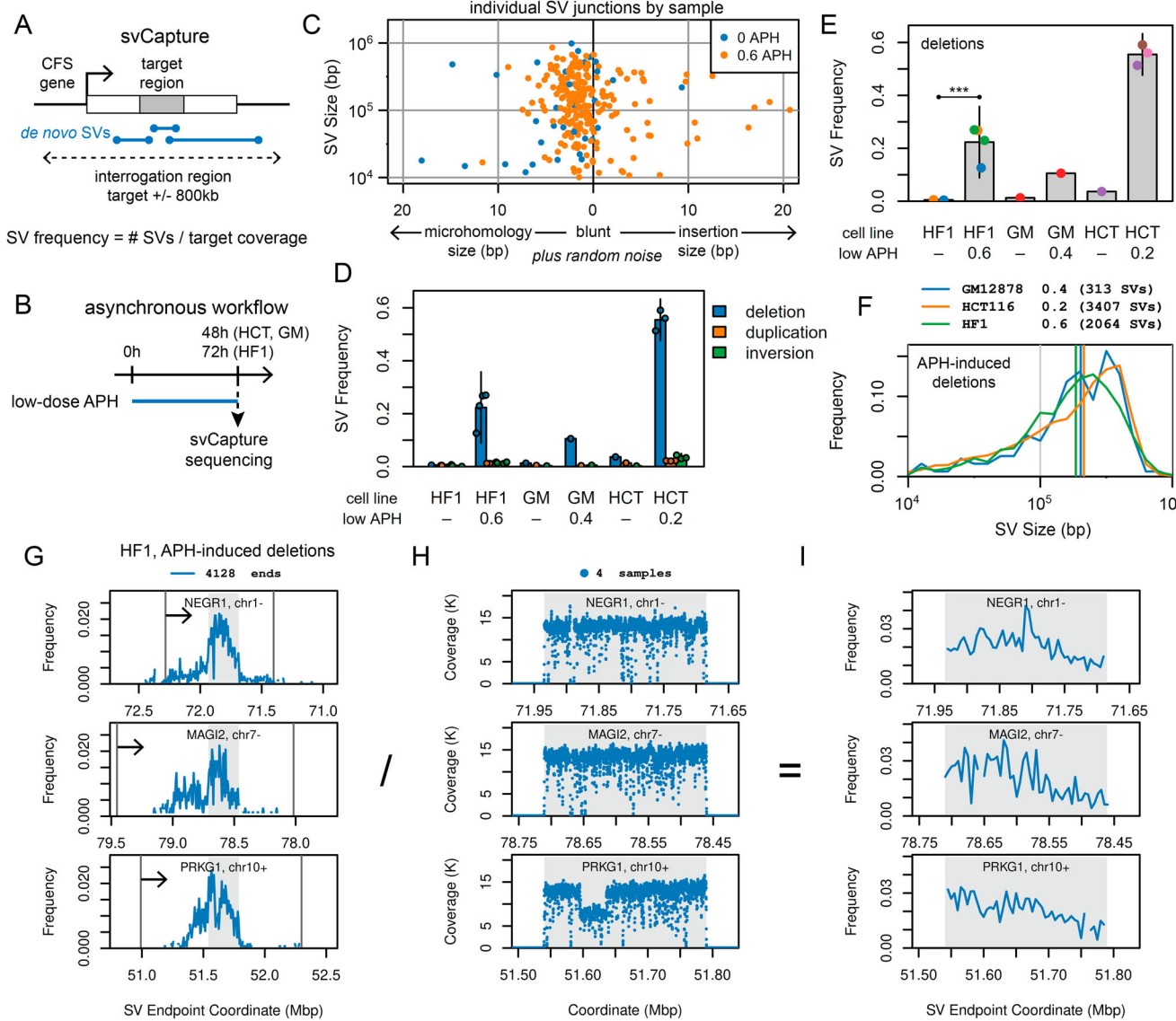

**Fig. 1 | APH-induced SV junctions arise throughout large CFS genes in asynchronous normal and cancer cell lines. A** svCapture sequencing was targeted near CFS gene centers to detect SV junctions with at least one breakpoint in a target region. **B** Timeline of asynchronous cell experiments, with SV induction by low-dose APH. **C** Example svCapture junction analysis from paired HF1 fibroblast samples shows the relationship between the sizes of SVs and their microhomologies, blunt ends, and insertions (see Methods). Each point is one intra-chromosomal, single-molecule SV between 10 kb and 1.2 Mb. SVs are plotted in random order with a small amount of random noise added to the X-axis to aid in visualization of all data points. There are fewer control/blue points ($n = 33$) vs. APH-treated points ($n = 222$). **D** Induced SVs are strongly biased to deletions. APH doses are in µM. SV frequency is the observed junction count divided by the target region fold-coverage. HF1, UM-HF1; GM, GM12878; HCT, HCT116. Independent biological replicate (total SV) numbers by cell line/APH are: HF1/-, 2 (55); HF1/ + , 4 (2295); GM/-, 1 (64); GM/ + , 1 (338); HCT/-, 1 (73); HCT/ + ; 3 (3757). Error bars are mean +/− 2 SD of two or more replicates. **E** Induced deletion SV frequency. Each point aggregates all SVs from one independent biological replicate, e.g., from one set of colored junctions in panel C. Sample point colors denote shared experimental batches handled together. Replicate (SV) numbers by cell line/APH are: HF1/-, 2 (22); HF1/ + ,

4 (2064); GM/-, 1 (42); GM/ + , 1 (313); HCT/-, 1 (51); HCT/ + ; 3 (3407). Error bars are mean +/− 2 SD of two or more replicates. Intergroup comparisons were made using a generalized linear regression model based on an over-dispersed Poisson, i.e., the negative binomial distribution, sampling SV junctions from sequenced haplotypes (see Methods). Throughout, two-tailed $p$-values of selected intergroup comparisons are marked as ns, not significant; *, $p <= 0.01$; **, $p <= 0.001$; ***, $p <= 0.0001$. HF1 $p$-value = 3.56e-63. **F** APH-induced deletion size distributions by cell line for the same data sources as in (**E**). Colored vertical lines are median SV sizes.
**G** Distribution of APH-induced deletion breakpoints in HF1 cells for the same data sources as in (**E**), which extend beyond the capture targets (shaded) but less than the limits of the interrogation regions (plot width) or gene spans (vertical lines, transcribed left to right, arrow). Data are aggregated in 5 kb bins. **H** High net HF1 target region read coverage in 100 bp bins for the same data sources as in G, the normalization denominator for panel (**I**). Low-coverage bins reflect reduced capture probe density and/or sequencing efficiency. *PRKG1* carried a clonal deletion in the HF1 cells under study at 51.595−51.645 Mb. **I** HF1 deletion breakpoint distributions in target regions for the same data sources as in G after normalizing to read coverage >=500, showing non-focal accumulation of SV junctions. Source data are provided as a Source Data file.

know what fraction of SVs observed without APH treatment are library artifacts vs. rare background events, but SVs accumulated above that background must have arisen during an experiment and provide a rich signal of dozens to hundreds of sequenced SV junctions per replicate sample (Fig. 1C and E).

## Many SVs induced at CFSs by aphidicolin stress form after passage into M-phase

A motivation in developing single-molecule svCapture was to determine when SVs form at CFS genes relative to replication. Replication stress occurs in S-phase, but we reasoned that subsequent junction

formation could occur in S concurrent with fork failure, in G2, in M associated with MiDAS or other processes, or in the next G1 associated with 53BP1 foci[42]. We, therefore, purified timed, flow-sorted cells and used real-time svCapture to assess when in the cell cycle APH-induced SV junctions formed. This study addresses the S, G2, and M-phases.

Our first experimental paradigm matched that used to study MiDAS[23]. HCT116 cells were treated with APH and synchronized at the G2-M boundary using the CDK1 inhibitor RO3306 (Fig. 2A). Treatment timing helped ensure that arrested cells had experienced APH-induced replication stress in the prior S-phase. G2 (4 N DNA content, phospho-histone H3[pH3] negative) and M-phase (4 N, pH3 positive) cells were harvested by flow cytometry prior to or 3 h after release from RO3306 arrest, respectively (Fig. 2C and Supplementary Fig. 3A–B). svCapture revealed a small but statistically significant increase in deletion SV formation in G2 cells treated with APH in the prior S-phase (Fig. 2D). However, deletion yield increased consistently by an average of 4.6-fold in M relative to G2-phase cells (Fig. 2D). Because cells were held in colchicine and flow sorted, they could not have passed into the next G1, indicating greater SV formation in M as compared to G2-phase. SVs that formed in M-phase had similar properties to those from bulk asynchronous cultures, including a large median size, a bias toward deletions, and a lower rate of induced duplications and inversions with distinct breakpoint distributions (Supplementary Fig. 4A–C).

To ensure RO3306 was not influencing results, e.g., by altering replication dynamics or inhibiting replication-associated repair in G2[43,44], we modified the workflow by either (i) harvesting G2 cells after release from RO3306 (Fig. 2B) or (ii) omitting RO3306 and using extended flow sorting to collect sufficient S, G2 and M-phase cells from asynchronous cultures (Fig. 2E and Supplementary Fig. 3C). In all cases, svCapture revealed a significant ~5-fold increase in deletion SV yield in M relative to G2-phase (Fig. 2D and F). To explore whether G2 cells were less capable of SV formation because they were dying, we assessed cell death by monitoring cleaved caspase 3 relative to a positive control treated with etoposide (Supplementary Fig. 3D). G2-phase HCT116 cells did not show an excess of cleaved caspase 3 despite being substantially less likely to have formed SV junctions than pH3-positive M-phase cells.

## MiDAS is not required for SV formation at CFSs under aphidicolin stress

Preferential SV formation at CFSs in M-phase might suggest that SV junctions are created by error-prone MiDAS, which occurs preferentially in large, actively transcribed genes, including CFS genes[23–25]. To test this possibility, we followed established protocols for inhibiting MiDAS by adding high-dose (2 μM) APH upon release from RO3306 arrest (Fig. 2G)[23]. To ensure MiDAS suppression, we added EdU to parallel cultures after RO3306 release and examined M-phase-specific EdU foci by fluorescence microscopy (Fig. 2G). High-dose APH abrogated MiDAS-associated EdU focus formation induced by low-dose APH (Fig. 2H–I). Interestingly, in contrast to some prior reports[23], most MiDAS M-phase EdU foci induced by low-dose APH exposure in S-phase were doublets with signal on both chromatids consistent with semi-conservative DNA replication (Fig. 2J), not the singlet foci restricted to one sister chromatid taken as evidence for conservative replication[23]. Most importantly, MiDAS inhibition by high-dose APH in M-phase HCT116 cells did not suppress deletion SV formation (Fig. 2K), a finding supported by a single-replicate experiment in GM12878 cells (Supplementary Fig. 4D).

## MiDAS inhibition does not affect CFS expression in HCT116 or GM12878 cells

SV formation and CFS expression are different manifestations of replication stress at CFS loci[18]. MiDAS inhibition was reported to greatly reduce total gaps and breaks and specific CFS expression in U2OS cells and to generate CFS-associated ultrafine bridges and

increased nondisjunction of chromosomes 3 and 16, which contain the FRA3B and FRA6D loci, in U2OS cells and MRC5 human fetal lung fibroblasts[23]. Because we found no effect of MiDAS inhibition on FRA3B or FRA16D-associated SV formation in HCT116 or GM12878 cells, we explored the relationship between MiDAS, chromosome breakage, and specific CFS expression in those cell lines. We first determined the optimal timing of high-dose APH (2uM) for MiDAS inhibition as determined by EdU foci formation in M-phase in unsynchronized cultures (Fig. 3A). In HCT116 cells, a 1-2 h high-APH treatment before chromosome harvest eliminated MiDAS (Fig. 3B). In contrast to HCT116 cells synchronized with RO3306 (Fig. 2H), there were a small number of foci seen in HCT116 cells treated with high APH for 3 h, presumably representing cells entering mitosis from early G2 or S-phase (Fig. 3B). For GM12878 cells, a 1 h high-APH treatment eliminated MiDAS, with low levels of EdU foci appearing in cells treated for 2 h (Fig. 3B). Based on these results, we used 1 h and 2 h high-APH treatments for cytogenetic analyses with both cell types. As with SV formation, we did not observe a decrease in APH-induced CFS expression or in total gaps and breaks in either HCT116 or GM12878 cells upon MiDAS inhibition with high-dose APH (Fig. 3C–F) and observed an increase in HCT116 breaks with increasing high-APH treatment time.

## SVs induced by aphidicolin stress have TMEJ-like junction profiles

Our results support SV formation at large CFS genes by a pathway activated in M-phase other than MiDAS. To inform candidate alternative(s) we performed a detailed analysis of an extensive database of 11,641 de novo deletion junctions sequenced to base-pair resolution from asynchronous and M-phase cells with unperturbed DNA repair. Short-read sequencing cannot reveal SVs created by HR, but prior microarray work never suggested events of that class[17]. Instead, both microarray and svCapture data support junctions strongly consistent with DSB end joining, where paired breakpoint positions in SV alleles revealed a range of microhomology usage, blunt joints, and de novo base insertions (Fig. 4A–C). The distribution of these junction classes, characterized by a prominent peak of 2 bp microhomology and a substantial minority of de novo insertions, was strikingly reproducible across APH treatment conditions (Fig. 4C), cell line and cell harvest workflow, cell cycle phase, SV type, and MiDAS status (Supplementary Fig. 5A–D).

We characterized junction insertions in detail because they are a signature of TMEJ[45–47], an end-joining pathway recently implicated in mitotic DSB repair[32–34]. Specifically, inserted bases are sometimes copied from template sequences near a DSB. The inferred repair process involves synthesis initiated from a priming microhomology flanking the inserted bases and eventual cross-DSB annealing via a second resolving microhomology on the opposite flank (Fig. 4B)[31,46]. Three TMEJ insertion classes with varying template orientations have been described, referred to here as foldback (also called inverse), cross-junction synthesis (also called direct), and a more complex and rarer strand-switching mechanism (Supplementary Fig. 6A–C)[46].

We searched for insertion templates 500 bp upstream and downstream of each reference genome breakpoint (Fig. 4B), requiring at least 7 bp template spans to promoted specificity. We found a significantly higher fraction of templates than expected by random chance across all insertion sizes from 1 to 20 bp (Fig. 4D and E). However, we did not find templates for most insertions and the hit rate decreased with insertion size, possibly due to a correlated increase in the frequency of untemplated or multi-template events.

A highly informative footprint emerged from the 1099 identified templates that featured a peak of both priming and resolving micro-homology lengths at 2 to 3 bp (Supplementary Fig. 5E) and a net total template size, including flanking microhomologies, of typically less than 10 bp (Supplementary Fig. 5F). Templates were almost always found on the retained side of genomic SV breakpoints (Fig. 4F),

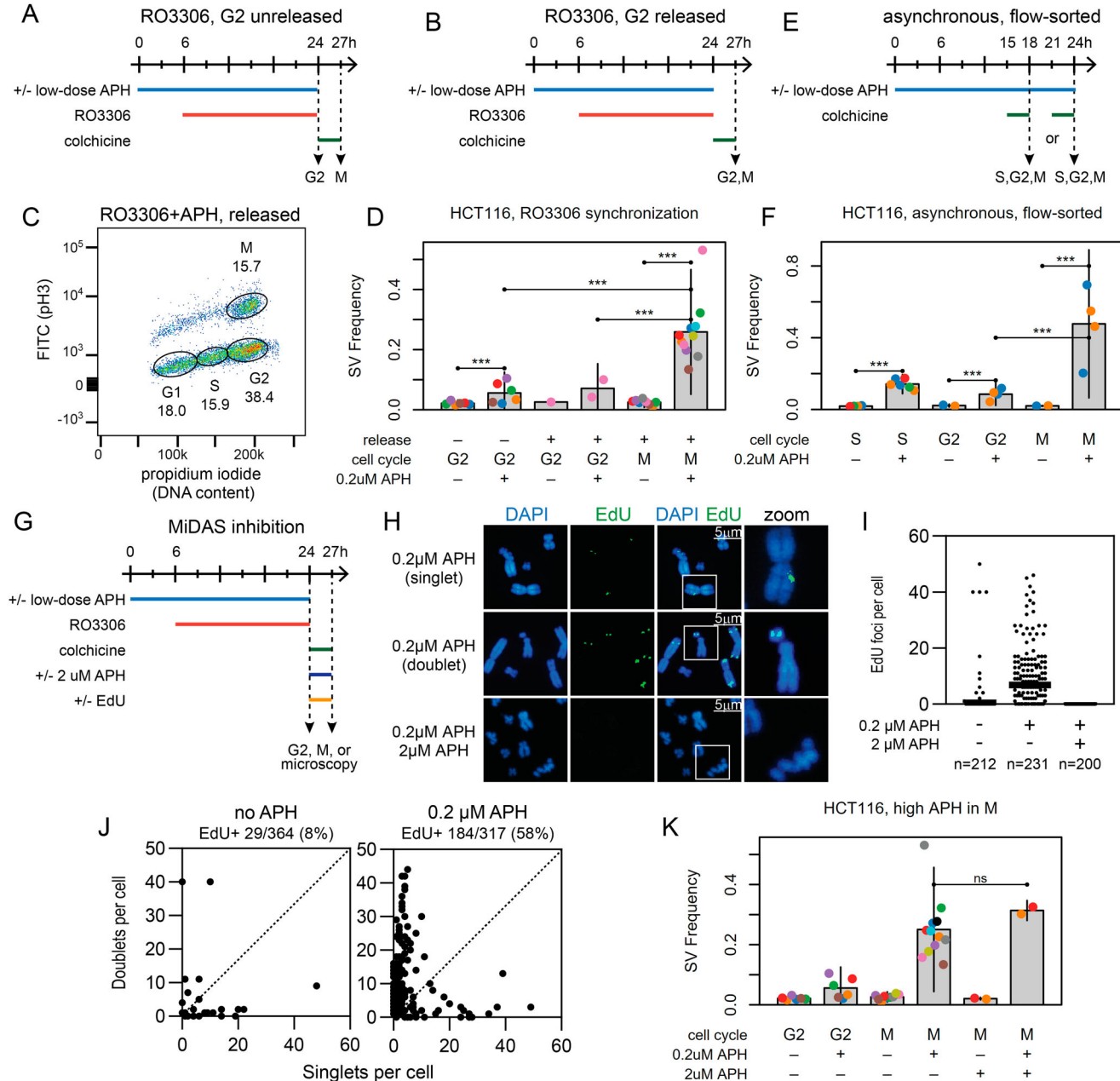

**Fig. 2 | APH induces SV junction formation during mitosis but independently of MiDAS. A**, **B** Two synchronization paradigms used to harvest APH-treated cells in different, timed phases of the cell cycle, where G2 cells were harvested before and after release from RO3306 arrest, respectively. **C** Example flow sorting of HCT116 cells after release from RO3006. x-axis, DNA content; y-axis, pH3. Gated cell fractions in cell cycle stages G1, S, G2, and M are shown. **D** SV frequencies from synchronized HCT116 cells harvested in the indicated cell cycle phases, showing increased APH-induced SV yield when cells passed from G2 to M. Points aggregate sequenced SVs from one independent biological replicate. Point colors denote experimental batches. Error bars are mean +/− 2 SD of two or more replicates. Replicate (SV) numbers by release/cell cycle/APH are: -/G2/-, 6 (241); -/G2/ + , 6 (555); +/G2/-, 1 (54); +/G2/ + , 2 (263); +/M/-, 9 (492); +/M/ + , 12 (6286). *P*-values from two-sided negative binomial generalized linear model: -/G2/- vs. -/G2/ + , 7.80e-10; -/G2/+ vs. +/M/ + , 8.12e-21; +/G2/+ vs. +/M/ + , 3.40e-13; +/M/- vs. +/M/ + , 4.67e-67. *, *p* <= 0.01; **, *p* <= 0.001; ***, *p* <= 0.0001. **E** Timeline of experiments where cells in different cell cycle phases were flow sorted from asynchronous cultures. Colchicine improved M-phase cell yield and prevented re-entry into G1/S. **F** SV frequencies from HCT116 cells using the paradigm in E, with M-phase cells again showing higher SV yield. Plot elements and statistics are the same as D. Independent biological replicate (SV) numbers by cell cycle/APH are: S/-, 4 (145); S/ + 6 (1672); G2/− 2 (78);

G2/ + 4 (546); M/− 2 (81); M/ + 4 (3680). *P*-values are: S/- vs. S/ + , 9.35e-89; G2/- vs. G2/ + , 9.93e-09; M/- vs. M/ + , 1.87e-22; G2/+ vs. M/ + , 1.88e-10. **G** Timeline of experiments where high-dose APH (2 μM) was added at release from RO3306 to inhibit MiDAS. EdU was added at RO3306 release only when visualizing MiDAS foci. **H** Example images of EdU foci scored in I and J, visualized in M-phase, showing singlet and doublet foci and the absence of foci with high-dose APH. Scale bar (white line): 5 μm. **I** Summary of EdU focus counts per metaphase from two independent biological experiments. Number of metaphase cells analyzed: no APH, 212; 0.2uM APH, 231; 0.2uM + 2uM APH, 200. **J** Comparison of EdU singlet and doublet focus yield between untreated and low-dose APH-treated cells over three independent biological experiments. Each point is one metaphase cell with at least one EdU focus, stratified by its singlet (x-axis) vs. doublet (y-axis) focus count. Number of EdU-positive metaphase cells/total metaphase cells analyzed: no APH, 29/364; 0.2uM APH, 184/317. **K** SV frequencies from HCT116 cells using the paradigm in G, with G2 cells harvested before RO3306 release, where suppression of MiDAS by high-dose APH in M had no impact on SV yield. Plot elements and statistics are the same as D. Independent biological replicate (SV) numbers by cell cycle/low APH/ high APH are: G2/-/-, 6 (241); G2/ + /-, 6 (555); M/-/-, 9 (492); M/ + /-, 12 (6286); M/-/ + , 2 (86); M/ + / + , 2 (1239). *P*-values by: M/+/- vs. M/ + / + , 0.109. ns: not significant, *p* > 0.01. Source data are provided as a Source Data file.

suggesting the search ensues after a DSB separates replicated/retained DNA from unreplicated/lost DNA. There was roughly equal utilization of foldback and cross-junction synthesis on either side of the SV junction, suggesting a random search (Fig. 4F). However, the search

was strongly restricted in distance from the junction, with a strong peak of template bases within 20 bp of breakpoints (Fig. 4F, G). Rare templates found at greater distances have increased likelihood of being random sequence matches (Supplementary Fig. 8A).

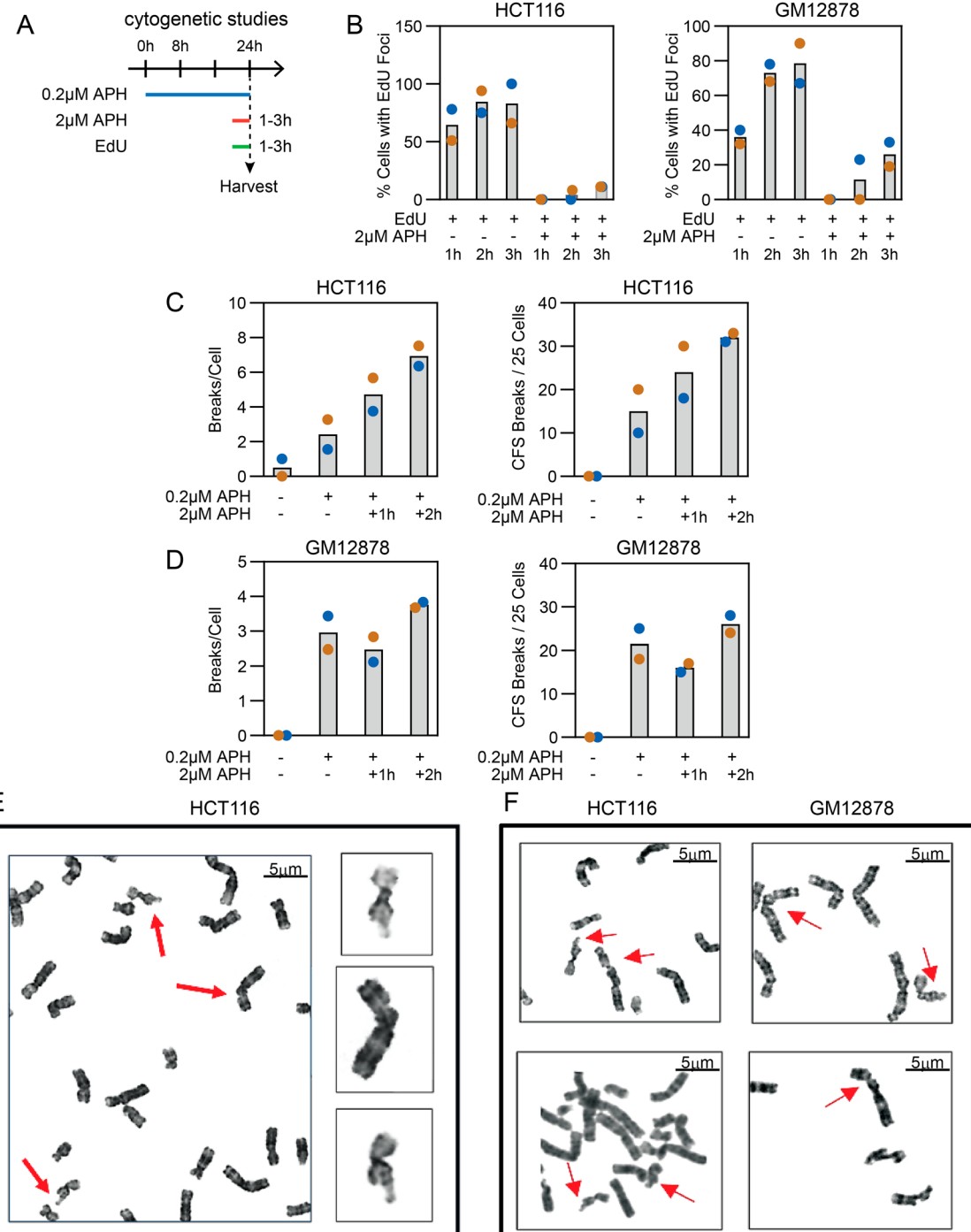

**Fig. 3 | APH-induced chromosome gaps and breaks and CFS expression do not depend on MiDAS. A** Timeline of experiments where high-dose APH (2 μM) and EdU were added at 1 h, 2 h, or 3 h prior to chromosome harvest to inhibit MiDAS and for visualizing MiDAS foci, respectively. **B** Comparison of EdU focus yield between untreated and high-dose APH-treated HCT116 and GM12878 cells. Gray bars are the mean of two independent biological replicates (orange and blue sample points). Number of HCT116 metaphase cells scored: 1 hr EdU 104 + 92; 2 hr EdU, 88 + 103; 3 hr EdU, 110 + 96; 1 hr EdU+2uMAPH, 109 + 100; 2 hr EdU+2uMAPH, 63 + 89; 3 hr EdU+2uM APH, 78 + 92. Number of GM12878 metaphase cells: 1 hr EdU, 59 + 102; 2 hr EdU, 59 + 88; 3 hr EdU, 41 + 92; 1 hr EdU+2uMAPH, 21 + 104; 2 hr EdU+2uMAPH,

29 + 83; 3 hr EdU+2uM APH, 21 + 77. **C, D** Total gaps and breaks (left) and CFS expression (right) in HCT116 and GM12878 cells, respectively, with and without high-dose APH to suppress MiDAS. Chromosome were analyzed in 25 metaphases from each of two biological replicates for each cell line. Gray bars, mean. **E** FRA3B and FRA116D CFS gaps/breaks (arrows) in a representative HCT116 cell treated with high-dose APH for 2 h from the set of experiments described in C. Scale bar (black line), 5 μm. **F** Additional examples of FRA3B and FRA16D CFS gaps/breaks (arrows) in HCT116 and GM12878 cells treated with high-dose APH for 2 h or 1 h, respectively, from the set of experiments described in (**C**) and (**D**). Scale bar (black line), 5 μm. Source data are provided as a Source Data file.

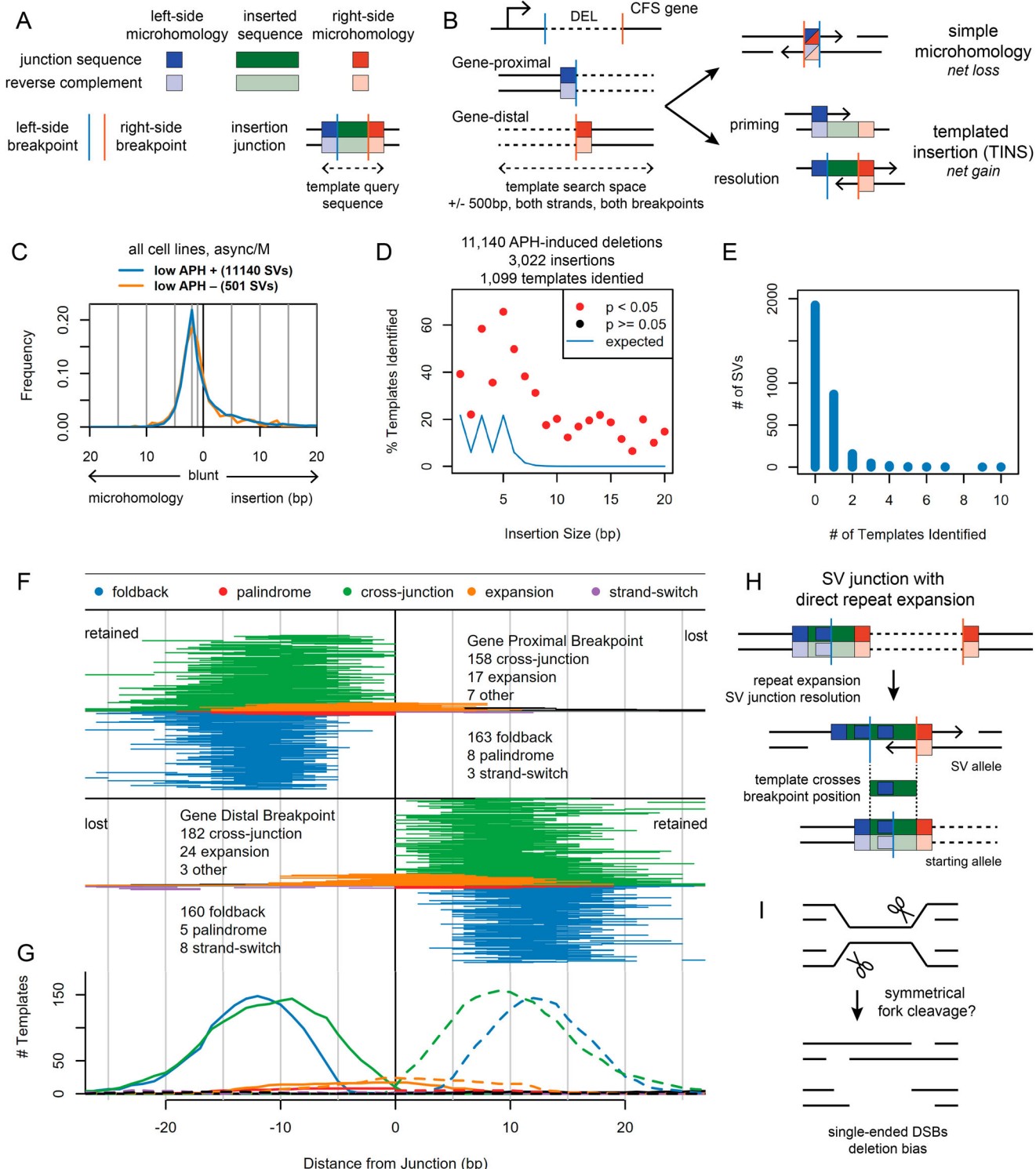

### Implications of insertion template patterns for fork cleavage leading to SVs

The mechanistic specificity of insertion templates was supported by a shift of foldback template bases nearest the junction toward positions 4 bp further away from the junction as compared to cross-junction synthesis (Fig. 4G and Supplementary Fig. 8B). This shift is consistent with the need for hairpin formation during foldback synthesis in most inverse insertions, which precludes the placement of priming micro-homologies at the extreme DSB terminus. However, we did observe a rare class of inverse insertions occurring at short palindromic sequences (Fig. 4F and Supplementary Fig. 6E). Because the palindrome bases could not all anneal in a hairpin, these insertions, and perhaps some that we describe as foldback events, may have occurred by an alternative bimolecular synthesis model that invokes the sister chromatid as template (Supplementary Fig. 6D).

Surprisingly, we found some insertion templates on the top gen-ome strand relative to a deletion that appeared to cross from the retained into the lost portion of the corresponding genomic break-point (orange lines in Fig. 4F). The events were characterized by short, often interrupted, direct repeats that were expanded from two to three repeat units in the final junction, so we refer to them as expansion insertions (Fig. 4H). In Supplementary Fig. 7, we draw two

**Fig. 4 | Junction analysis of >11 K de novo deletions implicates TMEJ repair of DSBs created by replication fork cleavage. A** Guide to drawing conventions used in this figure and models in supplemental figures. **B** SVs are oriented to transcription of the relevant CFS gene. Gene-proximal and gene-distal breakpoints can be paired via microhomologies that remove base pairs relative to the initial DSB ends, by blunt joints (not drawn), or by the insertion of novel bases, which might arise by template copying with sequential use of priming and resolving microhomologies. **C** Distribution of deletion junction microhomology and insertion sizes, i.e., breakpoint offsets, from all cell lines harvested asynchronously or in M-phase, stratified by APH induction. The microhomology peak is at 2 bp. Independent biological replicates (SVs) by APH: -, 14 (501); +, 23 (11140). **D** Yield of identified templates for the APH-induced samples and deletion SVs plotted in (**C**), stratified by insert size. 1099 of 3022 insertion-containing SVs analyzed had putative insertion templates identified. Smaller insertions were required to have more bases of flanking microhomology in candidate templates to maintain search specificity (see Methods), leading to the non-monotonic blue random expectation line. All insertion sizes showed significant enrichment of identified templates ($p <= 0.05$, red), as determined by a one-tailed assessment of the binomial distribution that asked

whether the fraction of identified templates for that size exceeded the number expected based on random Poisson sampling of bases (see Methods). Exact p-values are provided in a Source Data file. **E** Number of templates identified for insertion SVs from the data sources in (**D**) within 500 bp of junction breakpoints. 2788 of 3022 (92%) of insertion SVs had zero or one template identified, establishing search specificity. **F** Pileup of identified insertion template locations for the data sources in (**D**). The plot is oriented like panel B with respect to the genomic DNA regions surrounding the two breakpoints, *i.e.*, most templates were found in retained breakpoint segments. 738 templates are plotted in total. **G** Histogram of the bases contributing to insertion templates plotted in (**F**), emphasizing offset of foldback relative to cross-junction templates. Solid lines, left breakpoint; dashed lines, right breakpoint. Panels **F** and **G** share an X axis. **H** The recurrent structure of expansion-class insertions. See Supplementary Fig. 7 for models of how expansion might occur so that templates appear to cross into the deleted side of inferred breakpoint positions. **I** General model by which single-ended DSBs created at replication forks lead to SV formation mechanisms suggested by CFS SV junction analysis. Source data are provided as a Source Data file.

parsimonious models for these events (others may be possible). Importantly, designations of retained and lost DNA are relative to sequence alignment breakpoint positions, not the source DSB termini whose structure we do not know. For expansion junctions, the algorithm calls an insertion with a breakpoint upstream of the added repeat unit, which may or may not correspond to the starting DSB end. In one model, the leading strand template is cleaved and used as the template for fill-in synthesis of the resulting 5' overhanging DSB, where strand slippage would create the observed expansion (Supplementary Fig. 7). An alternative model invokes cleavage of the lagging strand template to yield a 3' overhanging DSB that again uses the leading strand template, now within a daughter strand gap, to create the expansion insertion (Supplementary Fig. 7). Expansion insertions have not been described for TMEJ occurring at CRISPR/Cas9-mediated DSBs[31,46], possibly because the mechanisms invoke either 5' overhanging DSBs or bimolecular reactions at forks (Fig. 4I) that do not apply to Cas9 blunt ends at simple DSBs.

## Chemical POLQ inhibition and POLQ knockout differentially impact SV formation

Because our junction analysis implicated TMEJ as a possible mechanism of SV formation at large CFS genes, we modified asynchronous and M-phase svCapture workflows to incorporate chemical POLQ inhibitors (Fig. 5A, B) and CRISPR-mediated *POLQ* knockout (KO) cell clones. We validated TMEJ loss using a published assay based on PCR detection of intracellular joining of transfected oligonucleotides (Fig. 5C and Supplementary Fig. 9)[48].

Results consistently supported a role for POLQ in SV formation at CFSs but varied by cell type and method with different degrees of SV loss. Asynchronous *POLQ*[-/-] HF1 and HCT116 cells each showed a reproducible, significant, but partial loss of deletions as well as other types of SVs relative to wild-type (Fig. 5D and E and Supplementary Fig. 10A and B). Deletion SV reduction was again partial in HCT116 asynchronous or M-phase cells treated with the POLQ inhibitors ART558[49] or novobiocin (NVB)[50,51] (Fig. 5E, F). In contrast, APH-treated *POLQ*[-/-] HCT116 M-phase cells showed baseline levels of SV formation with no apparent induction by APH (Fig. 5F and Supplementary Fig. 10C).

Because SV frequency alone might not reveal the full role of TMEJ if another repair pathway could partially replace it, we analyzed properties of residual SVs from cells with impaired TMEJ. SV sizes were similar regardless of POLQ status (Supplementary Fig. 10D–F). In contrast, we observed shifts in junction distributions comprising changes in microhomology lengths and insertion frequencies. *POLQ*[-/-] cells across all cell lines and workflows showed a near absence of insertions >=3 bp and a shift in peak microhomology length from 2 bp

to 1 bp (Fig. 5G–I and Supplementary Fig. 10G–I). In contrast, POLQ chemical inhibition did not substantially reduce insertions while the shift toward shorter microhomologies sometimes remained apparent (Fig. 5H–I and Supplementary Fig. 10H–I). We found templates for insertions detected in ART558 and NVB-treated samples much like uninhibited cells, although with a higher relative rate of expansion insertions (Supplementary Fig. 11). These results were supported by a single experimental replicate in GM12878 cells (Supplementary Fig. 10J–L).

Fig. 5J shows how APH-induced POLQ-proficient, POLQ-inhibited, and *POLQ*[-/-] samples group with respect to the insertion frequency and average microhomology lengths of their deletion SVs. The dynamics of these different manipulations must be carefully considered when interpreting results (Fig. 5K). POLQ protein loss is distinct from its chemical inhibition, which may only partially inhibit enzymatic activity and/or permit structural roles to be fulfilled. Moreover, inhibitors were used transiently whereas KO cells lacked POLQ from the time they were cloned. SVs that arose as background events before an experiment would form in POLQ proficient vs. deficient states for inhibition vs. KO, respectively, and these background events become a larger fraction of detected SVs as APH induction decreases (Fig. 5K).

## TMEJ and NHEJ cooperate in SV formation in some asynchronous cells

Despite challenges comparing POLQ chemical inhibitors and KO clones, results above demonstrate that some SV formation at CFSs can occur without POLQ, especially in asynchronous cultures. To explore the interplay between TMEJ and NHEJ in different cell cycle stages, we added chemical inhibition of DNA-PKcs using NU7441[52] and CRISPR/Cas9 KO of DNA ligase IV gene *LIG4* (Supplementary Fig. 12A and B)[53,54]. svCapture deletion SV yield in asynchronous HF1 cells was not altered by NU7441 in either wild-type or *POLQ*[-/-] backgrounds (Fig. 6A). In contrast, NU7441 significantly decreased deletion yield relative to ART558 or *POLQ* KO in asynchronous HCT116 cells (Fig. 6B) and in GM12878 cells in a single replicate (Supplementary Fig. 12C). This synergy was especially apparent when ART558 was added to asynchronous cultures of *LIG4*[-/-] HCT116 cells, which abrogated APH-induced SV formation (Fig. 6B).

A possible confounder above is that loss of both TMEJ and NHEJ can impair cell growth in some contexts[55], although we did not observe excessive cell death. To restrict the timeframe during which cells were doubly deficient, we added NU7441 to M-phase HCT116 cells just before release from RO3306 and observed that NU7441 now had no incremental impact on SV formation (Fig. 6C). As noted above, *POLQ* KO alone was sufficient to abrogate APH-induced SV formation in M-phase HCT116 cells (Fig. 6C). Throughout, loss of NHEJ by either

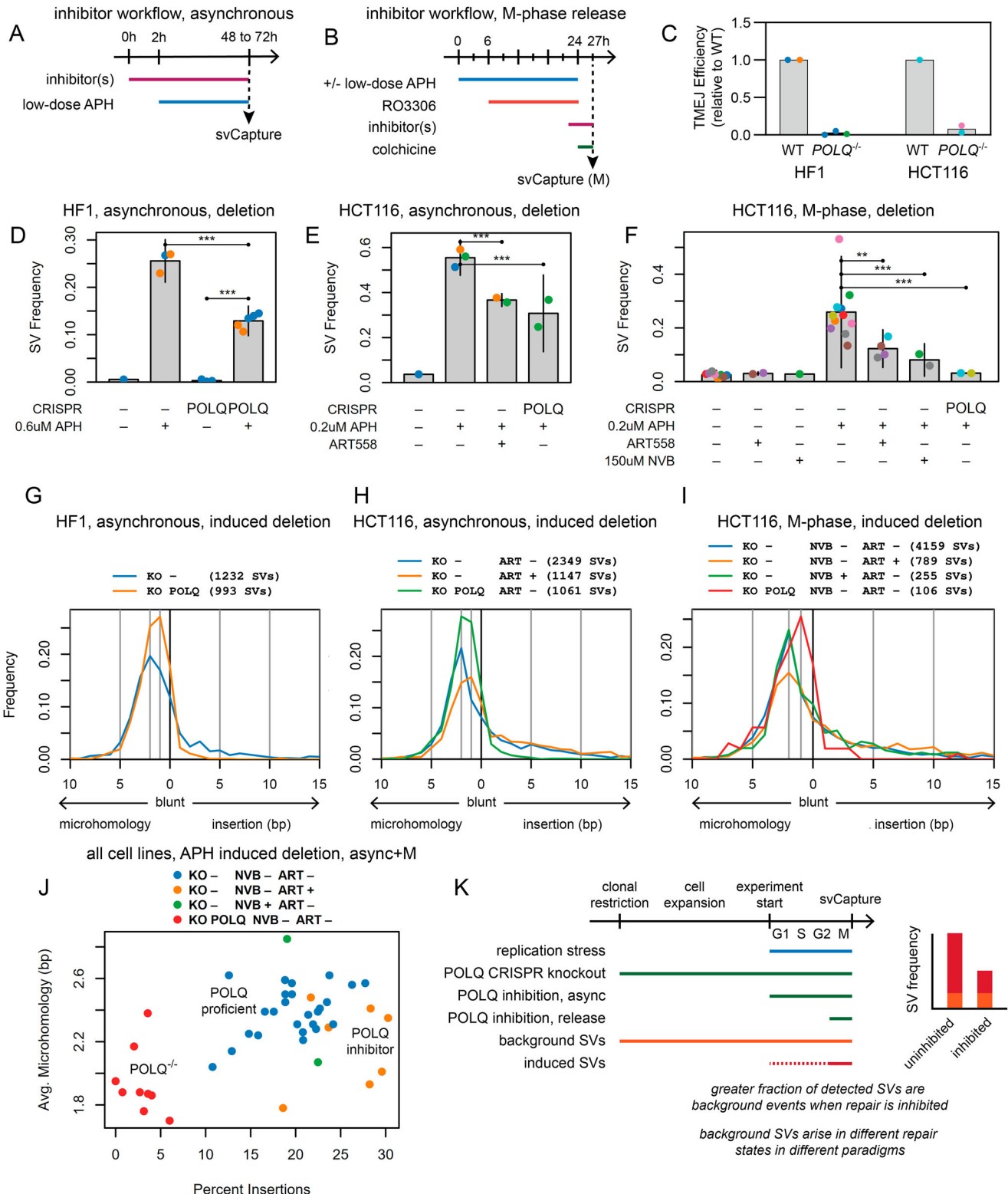

chemical inhibition or *LIG4* KO had no impact on junction insertion/microhomology distributions or insertion template locations (Fig. 6D and Supplementary Fig. 12D), consistent with these baseline junction properties being driven by TMEJ. Interestingly, NU7441 also had no impact on junction insertion/microhomology distributions in *POLQ*−/− cells (Figure Supplementary Fig. 12E).

Because non-TMEJ pathways might have a greater impact outside of M-phase, we repeated DNA repair manipulations above and harvested S-phase cells from asynchronous cultures treated with low-dose

APH for 18 hours (Fig. 6E). Although SV rates were lower in S as compared to M-phase, APH clearly induced some SV formation in S (Fig. 6F). Following results above, impairing NHEJ alone with *LIG4* KO had no impact on SV formation except for one of four replicates that we consider to be an outlier (Fig. 6F). In contrast, impairing TMEJ alone with *POLQ* KO partially reduced SV formation, while combined inhibition of both pathways by two experimental manipulations reduced SV formation to baseline levels (Fig. 6F). These results continue to reveal a larger role of TMEJ over NHEJ in CFS SV formation.

**Fig. 5 | POLQ inhibition and knockdown reduce SV formation in mitosis.**
**A, B** Modification of asynchronous and synchronization timelines, respectively, where DSB repair inhibitors were added prior to either APH addition or release into M-phase. **C** Loss of TMEJ in *POLQ*[-/-] HF1 and HCT116 cells as determined by joining of transfected oligonucleotides. Number of independent biological replicates: HF1 WT, 2; HF1 *POLQ*[-/-], 3; HCT116 WT, 1; HCT116 *POLQ*[-/-], 2. Sample point colors denote experimental batches. Gray bars, mean. **D** HF1 asynchronous deletion SV frequency with POLQ KO. Sample point colors denote experimental batches. Error bars are mean +/− 2 SD of two or more independent biological replicates. Replicate (total SV) numbers by CRISPR/APH are: -/-, 1 (12); -/+, 3 (1785); *POLQ*/-, 3 (20); *POLQ*/ +, 5 (1363). *P*-values from two-sided negative binomial generalized linear model: -/+ vs. *POLQ*/ +, 1.83e-47; *POLQ*/- vs. *POLQ*/ +, 4.90e-64. **, *p* <= 0.001; ***, *p* <= 0.0001. **E** HCT116 asynchronous deletion SV frequency with POLQ KO and inhibition by ART558. Plot labeling and statistics are the same as (**D**). Replicate (total SV) numbers by CRISPR/APH/ART558 are: -/-/-, 1 (51); -/+/-, 3 (3407); -/+/+, 2 (1799); *POLQ*/ +/-, 2 (1505). *P*-values are: -/+/- vs. -/+/+, 1.89e-49;-/+/- vs. *POLQ*/ +/-, 6.09e-05.

**F** HCT116 M-phase deletion SV frequency with POLQ KO and inhibition by ART558 and novobiocin (NVB). Plot labeling and statistics are the same as **D**. Replicate (total SV) numbers by CRISPR/APH/ART558/NVB are: -/-/-/-, 8 (425); -/-/+/-, 2 (114); -/-/-/+, 1 (37); -/+/-/-, 11 (5975); -/+/+/-, 4 (1175); -/+/-/+, 2 (352); *POLQ*/ +/-/-, 2 (153). *P*-values are: -/+/-/- vs. -/+/+/-, 5.42e-04; -/+/-/- vs. -/+/-/+, 7.11e-10; -/+/-/- vs. *POLQ*/ +/-/-, 8.42e-28. **G–I** APH-induced junction insertion/microhomology size distributions for the data sources in **D** to **F**, respectively. SV counts are shown above plots for all sample groups. **J** Summary of deletion junction property distributions for asynchronous plus M-phase samples. Each point is one biological replicate across all cell lines. The x-axis is the percent of deletion junctions in a sample that had 2 to 15 bp insertions, the y-axis is the average microhomology length of junctions without insertions. Numbers of replicates by CRISPR/NVB/ART558 are: -/-/-, 25; -/+/-, 7; -/+/-, 2; *POLQ*/-/-, 9. **K** Differences between repair \ression paradigms with respect to the SV junctions that are ultimately sequenced. The SV frequency plot is conceptual and does not represent actual data. Source data are provided as a Source Data file.

## Discussion

This study addresses nonrecurrent SVs that arise in non-repetitive loci by mechanisms linked to negative interactions between transcription and replication. Unlike the early replicating genome, where these interactions include machinery collisions and R-loop formation[22,56–58], instability in the late replicating genome relates more to conflicts that cause unreplicated DNA to be propagated into M-phase[23–25]. The most unstable late-replicating loci are at the largest human CFS genes, where S-phase transcription impedes origin usage and completion of replication[22,41,59]. We show that some of the mechanisms evolved to resolve the resulting unreplicated DNA prior to cell division can create large non-recurrent SVs in at least some genomic loci.

We monitored SV junctions as they formed in cells using svCapture. svCapture[36] could detect timed de novo SV formation in part due to the high rate of CNV formation at transcribed CFS genes under replication stress[17] and in part because it offers single-molecule sensitivity[36]. Prior work examined features such as single-chromosome end-to-end fusions in Ku-deficient cells, radial formation in Fanconi anemia cells, and the persistence of DNA damage evident as DNA repair foci[33,60], but ours is a direct demonstration of intrachromosomal SV junction formation in M-phase, including large SVs spanning hundreds of kb. We characterized 48,362 such de novo SVs in controlled, prospective experiments with a combined 322,303-fold target coverage, which provides a uniquely powerful dataset for exploring junction formation mechanisms. Multiple observations affirm that we are measuring bona fide SV junctions. All aspects of CFS CNVs observed in microarray studies[12–18] were recapitulated by svCapture, but with much greater resolution and depth. Moreover, junction microhomology and templated insertions changed with cellular DNA repair capacity and must therefore comprise SVs created in cells.

Mathematical models[61] have shown that stochastic fork failure coupled with an inability to rescue replication via dormant origins can alone yield the central peak of SV junctions we see within CFS genes[17]. Consistently, no subregions in any of the CFS genes we studied acted as more highly localized SV hotspots. The large introns in CFS genes often have AT-rich sequences that have been invoked as important for CFS expression through the formation of focal flexibility peaks[38,39]. Biochemical studies of DNA polymerase progression through specific focal regions of CFS genes, especially sequences in *WWOX*, further showed they can impose mechanistic barriers to replication[62,63]. However, svCapture data linked CFS gene transcription to a diffuse distribution of de novo junctions. A limitation in making these conclusions is that extensive processing of DSBs could make SV distributions more diffuse than the underlying source lesions, whose structure cannot be determined from junction sequences.

CFSs replicate as late as M-phase[17,20,21,23,64], and the distribution of SV junctions we observed at CFS genes corresponds well to the location of MiDAS synthesis peaks observed under replication stress[24,25].

Thus, replication inhibition and the initiation of instability begin in S-phase but are not resolved until late in the cell cycle. We hypothesized SV formation would also occur in late G2 or M-phase coincident with completion of replication, which was confirmed by direct measurement that 80% of pre-G1 CFS SV junction formation occurs after cells pass into M-phase. Specifically, most SV junction formation occurred after the appearance of pH3 in chromatin coincident with MiDAS timing as previously reported[23]. Importantly, RO3306 was not artifactually suppressing SV formation in G2-phase secondary to CDK1 inhibition[43,44]. Colchicine addition and flow sorting further ensured that junctions reported here were not formed secondary to rupture of ultrafine bridges and progression into the next G1. However, the current work does not address the non-exclusive possibility that additional SV formation might occur in G1 associated with the appearance of 53BP1 bodies[23,42], especially in the absence of POLQ. Such studies will be complicated by the need to distinguish between SVs formed in M vs. G1 following a stressed S-phase, and by technical factors related to cell purity, but will be important for a complete understanding of SV formation timing.

Within M-phase, multiple possibilities existed for SV junction formation at CFSs. Because MiDAS is an error-prone form of conservative replication[23,26,65], and because MiDAS hotspots are found in CFS loci[23–25], it was a plausible mechanism for executing SV junction formation. However, MiDAS proved to not be required nor did suppressing it significantly increase SV junction formation or reduce CFS expression in GM12878 or HCT116 cells, contrary to prior studies with other cell types[23]. Thus, MiDAS appears to be a genome preserving pathway for completing synthesis of unreplicated DNA in mitosis, but MiDAS and SV formation do not appear to function as competing pathways. Importantly, the spans of unreplicated DNA passing into M-phase at CFS genes are likely exceptionally large and more numerous relative to expectations for the rest of the genome and under more typical cellular stresses than chemical inhibition with APH. MiDAS may never be fully effective in completing replication of hundreds of kb of CFS DNA, which might explain our data if SV formation could occur even after aborted MiDAS.

Detailed analysis of 33,279 fully sequenced SV junctions showed a remarkably reproducible junction pattern across SV types, cell cycle phases, and APH-induction status. In repair-proficient cells, that profile included a prominent peak at 2 bp microhomology and a long tail of ~20% de novo base insertions, a signature of TMEJ indicative of DSBs being a key intermediate in CFS SV formation[46,47,66]. Short-read sequencing has detection biases against longer insertions >20 bp but is fully able to reveal nonhomologous junctions with several bases of microhomology or insertion. Thus, the junction profile we observed is a reliable signature for the mechanism(s) catalyzing de novo SV junction formation at CFSs, especially in M-phase. SV junctions arising at CFS genes used insertion templates restricted to a window of

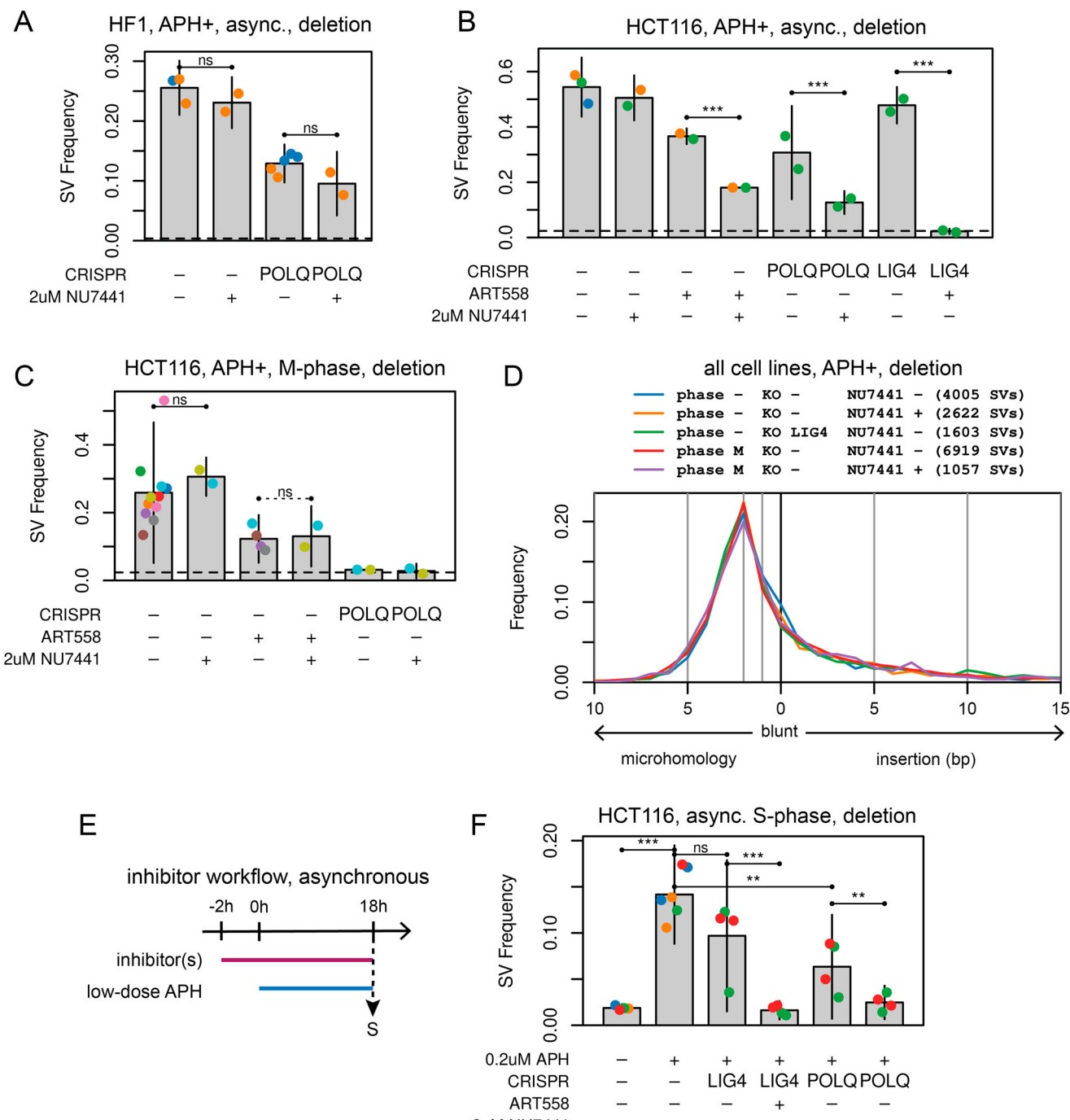

**Fig. 6 | Cell-cycle dependent interplay between TMEJ and NHEJ in SV formation at CFSs. A** Deletion SV frequency in asynchronous, APH-induced HF1 cells as a function of *POLQ* knockout and NHEJ inhibition with Nu7441. Sample point colors denote experimental batches. Error bars are mean +/− 2 SD of two or more independent biological replicates. Replicate (total SV) numbers by CRISPR/Nu7441 are: -/-, 3 (1785); -/+, 2 (1343); *POLQ*/-, 5 (1363); *POLQ*/ + , 2 (524). *P*-values from two-sided negative binomial generalized linear model: -/- vs. -/+, 0.180; *POLQ*/- vs. *POLQ*/ + , 0.119. ns, not significant; **, *p* <= 0.001; ***, *p* <= 0.0001. **B** Deletion SV frequency in asynchronous, APH-induced HCT116 cells as a function of *POLQ* and *LIG4* knockout, and inhibition of NHEJ with Nu7441 or TMEJ with ART558. Plot labeling and statistics are the same as A. Replicate (total SV) numbers by CRISPR/ART558/Nu7441 are: -/-/-, 3 (3407); -/-/+, 2 (2512); -/+/-, 2 (1799); -/+/+, 2 (848); *POLQ*/-/-, 2 (1505); *POLQ*/-/ + , 2 (668); *LIG4*/-/-, 2 (2322); *LIG4*/ +/-, 2 (116). *P*-values are: -/+/- vs. -/+/+, 5.59e-64; *POLQ*/-/- vs. *POLQ*/-/ + , 6.23e-08; *LIG4*/-/- vs. *LIG4*/ +/-, 8.89e-197. **C** Like A and B, now for HCT116 cells released into M-phase following RO3306 arrest. Plot labeling and statistics are the same as A. Replicate (total SV) numbers by CRISPR/ART558/Nu7441 are: -/-/-, 11 (5975); -/-/+, 2 (1473); -/+/-, 4 (1175); -/+/+, 2 (681); *POLQ*/-/-, 2

(153); *POLQ*/-/ + , 2 (148). *P*-values are: -/-/- vs. -/-/+, 0.376; -/+/- vs. -/+/+, 0.571. For clarity, panels A to C only show samples induced to form SVs with low-dose APH. Horizontal dashed lines indicate the cell-line-specific SV level consistently observed without APH addition. **D** Like Fig. 4C, showing no effect of NHEJ loss on junction microhomology and insertion profiles. Independent biological replicates (total SV) numbers by cell cycle/CRISPR/Nu7441 are: async(-)/-/-, 8 (4005); async(-)/-/+, 5 (2622); async(-)/*LIG4*/-, 2 (1603); M/-/-, 14 (6919); M/-/ + , 1 (1057). **E** Timeline for harvesting S-phase HCT116 cells from asynchronous cultures. **F** Deletion SV frequency in S-phase HCT116 cells as a function of *POLQ* and *LIG4* knockout, and inhibition of NHEJ with Nu7441 or TMEJ with ART558. Plot labeling and statistics are the same as A. Replicate (total SV) numbers by APH/CRISPR/ART558/Nu7441 are: -/-/-/-, 4 (145); +/-/-/-, 6 (1672); +/LIG4/-/-, 4 (841); +/LIG4/ + /-, 4 (139); +/POLQ/-/-, 4 (565); +/POLQ/-/ + , 4 (205). *P*-values are: -/-/-/- vs. +/-/-/-, 9.35e-89; +/-/-/- vs. +/LIG4/-/-, 5.96e-02; +/-/-/- vs. +/POLQ/-/-, 1.67e-04; +/LIG4/-/- vs. +/LIG4/ + /-, 1.95e-14; +/POLQ/-/- vs. +/POLQ/-/ + , 3.89e-04. Source data are provided as a Source Data file.

approximately 20 bp into the presumed DSB terminus. These insertion events were lost in *POLQ* KO, which establishes constraining parameters against which biochemical and structural studies of POLQ should be compared. Interestingly, templates were found for a minority of de novo insertions. We suspect this is mostly due to iterative copying of multiple templates and/or untemplated synthesis by POLQ[66,67] but cannot rule out other processes. We also cannot rule out non-TMEJ contributions of POLQ, e.g., its role in gap filling following BRCA1/2 loss or PARP inhibition[68], but the most parsimonious explanation of our data is that POLQ catalyzes junction formation over long distances through TMEJ. Studies of additional TMEJ factors will help support this conclusion. Even if gaps and associated processing by POLQ prove important in some modes of replication stress, resolution of lesions to the class of SV junctions we observed seems to require DSB end joining.

The best demonstration of a role for POLQ in SV junction formation was the loss of APH-induced M-phase SVs in *POLQ*⁻/⁻ HCT116 cells. That result is strongly consistent with recent studies identifying POLQ suppression by RAD52 and BRCA2 prior to M-phase[69] and its activation in M-phase by RHINO-mediated recruitment to DSBs and phosphorylation by PLK1[32–34]. Our results support that TMEJ acts an error-prone rescue pathway for dealing with unreplicated DNA in M-phase to prevent mitotic catastrophe. TMEJ has been shown to produce short 50-200 bp deletions in mutation accumulation experiments in *C. elegans*[70] and 5-50 bp deletions at Cas9-induced DSBs in mouse cells[71] but those small SVs are best modeled as occurring via resection of a single two-ended DSB. Our work establishes that POLQ can mediate M-phase formation of large, multi-lesion, spontaneous SVs >100 kb in mammalian cells.

Given the above, we were surprised by the partial effects of POLQ inhibitors on SV formation in multiple cell lines, including the widely used and more specific agent ART558[49,72]. POLQ inhibition did sometimes reduce microhomology lengths similarly to *POLQ* KO, arguing we effectively inhibited POLQ, but with a relative persistence of templated insertions. Because our experiments measured SV junction formation as the primary outcome, they may reveal a separation of POLQ functions[73]. ART558 binds to the polymerase catalytic domain of PolQ and inhibits its activity[49], whereas NVB inhibits the POLQ ATPase[50,51]. Neither removes the potential for POLQ acting in non-catalytic ways. Based on our results, microhomology use appears to be more impacted by alterations in POLQ catalytic activities whereas insertion formation strongly depends on structural functions. However, chemical inhibition of POLQ was transient whereas POLQ KO preceded the experimental window, which impacts the SVs that svCapture would detect as background events.

Despite evidence that TMEJ acts in M-phase as a source of large CFS SVs, that association is incomplete. Even with *POLQ* KO, some de novo SV formation remained in asynchronous cell cultures. This effect could not be explained by different M-phase behavior of cancerous and normal cells[74] because it was observed in both the HF1 and HCT116 cell lines. Some SV formation must be catalyzed by another end-joining pathway, at least as a backup to TMEJ. Importantly, in both the current work and our prior study of murine Xrcc4⁻/⁻ embryonic stem cells[75], we saw no impact of the loss of NHEJ alone on SV formation even in S-phase, identifying it as secondary to TMEJ. It is difficult to rule out that some SVs detected in S or G2 were formed in a prior M-phase but given that NHEJ is thought to be largely inactive in M-phase[76,77], it seems more likely that non-TMEJ pathways had a greater role in SV formation outside of M-phase. Of note, DNA polymerase lambda has been suggested to mediate end joining independently of NHEJ and TMEJ and might also play a role in SV formation[78].

Insertion templates that appeared to cross breakpoint junctions were unexpected but can be modeled as arising from repeat expansions created at the single-ended DSBs known to be formed in CFSs[79], presumably in M-phase by fork cleavage that activates replication rescue. Both the MUS81 and GEN1 structure-specific nucleases are required for CFS expression[80,81] and are excellent candidates for creating those source DSBs leading to SV formation. If the slippage expansion model is correct, it would implicate MUS81 as it is thought to cleave the leading strand template at stalled forks, the orientation that would yield 5' overhangs[82]. The sister-chromatid synthesis model might implicate GEN1 or another nuclease, but we currently favor leading strand cleavage in part because the locations of inverse templates appeared to favor unimolecular foldback synthesis over bimolecular sister synthesis. More work is required to address fork cleavage polarity leading to SV formation, but in either case, symmetrical cleavage of stalled forks has been proposed to lead to HR-independent SCEs[35,83] where DSB repair by TMEJ would obligatorily lead to deletion SVs corresponding to unreplicated DNA spans. It is less obvious how TMEJ would lead to duplications and inversions, which, although less frequent, were seen at CFSs and bore the hallmarks of TMEJ.

Our results provide key insights into the ways that error-prone replication rescue in M-phase can proceed from DSB formation to the creation of various types of SVs. It is not yet known to what extent the final resolution of replication-associated damage throughout the genome is deferred until M-phase, but MiDAS-associated propagation of unreplicated, R-loop-associated DNA into M-phase has been observed in *BRCA2* and *RAD51*-deficient cells and cells with cyclin E1 overexpression[57,84,85]. Moreover, we have observed replication stress-induced CNVs that mimic clinically important mammalian non-recurrent CNVs at numerous non-CFS loci using microarrays[17]. Taken together, it appears possible that some SVs in non-CFS loci may also be created in M-phase by TMEJ. Indeed, our results closely match prior descriptions of microhomologies and frequent insertions at SV junctions in both normal human CNVs[47,86,87] and SVs in cancers[88–90], implicating POLQ and mitotic TMEJ as potential mechanisms in mammalian SV mutagenesis. If true, the risk of TMEJ-mediated SV formation would likely be elevated in cancers and other cells that depend on TMEJ due to down-regulation of HR or NHEJ[91,92].

## Methods

### Cell Culture Models

**UM-HF1 fibroblasts.** UM-HF1 (abbreviated HF1 throughout) is a XY male, euploid, TERT-immortalized human foreskin-derived fibroblast cell line derived and maintained at the University of Michigan[17] subject to human data access restrictions. It has known CFSs/CNV hotspots at genes *PRKG1*, *NEGR1*, and *MAGI2* that provide excellent svCapture signal in a non-cancerous cell line[17,36], but HF1 cells are not easily synchronized for cell-cycle analysis. HF1 cells were cultured at 37 C with 5% $CO_2$ in Dulbecco's Modified Eagle Medium supplemented with 13% fetal bovine serum (FBS), 2mM L-glutamine, and 100 U/ml penicillin-streptomycin (Gibco).

**GM12878 lymphoblastoid cells.** GM12878, (Coriell, RRID CVCL_7526), is a highly studied XX female, euploid, EBV-immortalized human lymphoblastoid cell line generated as part of the HapMap Project. It has CFSs common to lymphoblastoid cells at genes *WWOX* and *FHIT* that allow direct comparison of CFSs and SVs in a suspension cell line. GM12878 cells were cultured at 37 C with 5% $CO_2$ in RPMI 1640 medium supplemented with 15% FBS, 2mM L-glutamine, and 100 U/ml penicillin-streptomycin.

**HCT116 colon cancer cells.** HCT116 (ATCC, RRID CVCL_0291) is a highly studied male, mismatch-repair deficient, human colon cancer cell line. This work established that genes *WWOX* and *FHIT* are SV hotspots in HCT116, consistent with CFS expression analysis, gene expression analysis using Bru-seq[93], and genomic analysis that showed baseline SVs in these genes[94]. HCT116 cells were cultured at 37 C with 5% $CO_2$ in McCoy's 5 A medium supplemented with 10% FBS, 2mM L-glutamine, and 100 U/ml penicillin-streptomycin.

## CRISPR-Cas9-mediated gene knockout

For CRISPR-Cas9 mediated knockout (KO) of *POLQ* or *LIG4* in HCT116 cells, single guide RNAs (sgRNAs, Supplementary Data 4) were designed using CHOPCHOP[95] and cloned into the PX459 plasmid, which carries the human U6 promoter, Cas9 gene, and puromycin resistance gene[96]. The plasmids were transfected into HCT116 cells using Lipofectamine 3000 (Invitrogen). For *POLQ* KO in HF1 fibroblasts, vector pLentiCRISPR v2 with integrated sgRNAs (GenScript) was transfected into HF1 cells by the University of Michigan Vector Core. Following transfection, cells were subjected to selection with 1 μg/ml puromyocin and KO clones were established by plating at low-density and isolating single colonies with cloning rings for expansion in multi-well dishes. Following clonal expansion, PCR with primers flanking the sgRNA binding sites was used to amplify the target alleles followed by Sanger sequencing. The resulting mixed allelic sequence traces were analyzed using Synthego ICE software[97]. Clones were preferred for further use when each of the two alleles yielded distinct frameshift mutations (Supplementary Data 4). Clones that yielded identical biallelic mutations were confirmed with ddPCR to establish a copy number of two for the mutant allele. *LIG4* KO clones were further validated with immunoblotting. The large protein size and low expression of POLQ resulted in inconclusive westerns, so *POLQ* KO clones were validated using the TMEJ assay described below. At least two independent clones of all mutated cell lines were frozen at early passage post-cloning for subsequent SV analysis.

## Replication stress induction and monitoring

The DNA polymerase inhibitor aphidicolin (APH, Sigma) was dissolved in DMSO at a stock concentration of 200 μM. For CNV and CFS induction, cells were cultured with APH as follows: HF1, 0.6 μM; GM12878, 0.4 μM; HCT116, 0.2 μM. These doses were empirically determined per cell line to be consistent with slowed but continued cell division and to produce approximately 2-5 chromosome gaps and breaks per cell in GM12878 cells. These concentrations of APH are defined as low-dose APH throughout. The duration of low-dose APH treatment and its timing relative to other cell manipulations varies is shown in timelines in relevant figures. High-dose APH (2 μM) and EdU (10 μM), to suppress MiDAS or reveal MiDAS foci, respectively, were added 1 to 3 h prior to harvest.

## Chromosome breaks and common fragile sites

CFSs and total gaps and breaks were scored on Giemsa-banded metaphase preparations following 24 h low-dose APH induction with or without the addition of high-dose APH as described above. Cells were harvested for chromosome preparations using standard conditions of a 20 to 45 min Colcemid treatment (50 ng/ml: Gibco) followed by a 15 min incubation in 0.075 M KCl hypotonic solution at 37 C and multiple changes of Carnoy's fixative (3:1 methanol:acetic acid). Fixed cells were dropped onto glass slides to generate metaphase spreads and slides were baked overnight at 60 C before Giemsa banding. For Giemsa banding, slides were dipped in water, treated with trypsin solution (0.0005% trypsin and 0.02% Tyrode's diluted in HBSS) for 50 s, followed by two rinses with 0.9% NaCl, stained in Giemsa staining solution (5% Giemsa in Gurr Buffer, pH 6.8) for 5 min, followed by two sequential rinses in water. Metaphase chromosomes were visualized using Zeiss Axiphot microscope and chromosome breaks and gaps were analyzed in 25-50 metaphases from each experimental sample.

## Cell synchronization and flow sorting

Cells were treated with low-dose APH for an initial 6 h and then arrested at the G2/M boundary by addition of 9 μM (HCT116) or 10 μM (GM12878) RO3306 (ApexBio) with continued APH for an additional 18 h. Parallel cultures were then either harvested for flow sorting of S and G2 fractions or washed three times with PBS and released into warm media containing 75 ng/ml colchicine for 3 h for flow sorting of

G2 and M fractions and also with 10 μM EdU for cells used to visualize MiDAS foci. For cells treated with novobiocin (NVB, 150 μM, Sigma), the drug was added together with low-dose APH and added back again after release from RO3306. ART558 (5 μM or 10 μM, MedChemExpress) or Nu7741 (2 μM, Fisher Scientific) were added 2 h prior to harvest for S and G2 fractions or RO3306 release and again added back to the media after release. High-dose (2 μM) APH was added for 3 h after cells were released from RO3306. When performing cell cycle analysis without RO3306, asynchronous HCT116 cells were treated for 24 h with low-dose aphidicolin. Three hours prior to harvest 100 ng/ml Colcemid was added to the media to enrich the M-phase population.

For flow cytometry, cells were harvested with trypsinization, collected in cold media, spun down (5 min, 500xg, 4 C), and fixed in 70% ethanol overnight at −20C at a concentration of 1×10⁶ to 2×10⁶ cells/ml. Cells were then washed with PBS, permeabilized with 0.25% Triton X-100 in PBS on ice for 15 min, spun down, and stained with phospho-histone H3 (pH3) antibody (Cell Signaling Technology) conjugated to Alexa fluor 488 (Invitrogen) at a 1:50 dilution in antibody staining buffer (0.5% bovine serum albumin (BSA) in PBS) for 1 h. Cells were washed twice with antibody staining buffer and stained with a solution of 100 μg/ml propidium iodide and 100 μg/ml RNASe. Samples were then submitted to the University of Michigan Flow Cytometry Core for collecting cell cycle fractions using a FACS Aria III (BD Bioscience) or Bigfoot Cell Sorter (ThermoFisher). Gating established that S-phase fractions had a DNA content between 2 N and 4 N and were pH3 negative, G2-phase fractions had 4 N DNA content and were pH3 negative, and M-phase fractions had 4 N DNA content and were pH3 positive. Flow sorting was continued until at least 200,000 cells had been collected from all target cell cycle fractions in a sample.

## MiDAS assessment

After initial cell harvesting, cells treated with EdU after release from RO3306 were prepared for metaphase analysis as described above for chromosome spreads. MiDAS activity was then assessed using Click-iT reaction and Alexa Fluor 488 azide (Invitrogen). Slides were first treated with 4% formaldehyde in PBS for 4 min, washed three times with PBS, and blocked with 3% BSA in PBS for 30 min. Permeabilization and the Click-iT reaction were performed according to the manufacturer's instructions. Slides were then washed with 3% BSA/0.5% Triton X-100 in PBS three times for 10 min per wash, rinsed with water and mounted with Prolong Gold DAPI Antifade mounting media (Sigma). Metaphase chromosomes were visualized using Zeiss Axiphot fluorescence microscope. Images were acquired using CellSens software. EdU quantification was done manually at the microscope, counting for every cell the number of EdU foci stratified by whether just one (singlet) or both (doublet) chromatids were labeled.

## TMEJ and NHEJ assays

To monitor the efficacy of chemical and genetic interventions intended to inhibit POLQ/TMEJ or NHEJ, we used an assay based on transfected oligonucleotides substrates subjected to intracellular end joining[48]. The substrates were prepared by the Dale Ramsden laboratory by annealing substrates in a buffer containing 10 mM Tris-HCL, pH 7.5, 100 mM NaCl, and 0.1 mM EDTA. 5 ng of NHEJ substrate or 500 ng of TMEJ substrate were then electroporated separately in a solution containing 500 ng pUC19 plasmid, 0.16 μl 2X PBS, 0.84 μl EB buffer and buffer R in a total of 10 μl into 200,000 cells using the Neon system with a single pulse of 1350 V (GM12878) or 1530 V (HCT116) for 20 ms. Prior to electroporation, cells were pretreated for 2 h with 10 μM ART558, 150 μM NVB, or 2uM Nu7441. After electroporation, cells were incubated in antibiotic-free media supplemented with the appropriate drug for another 30 min at 37 C, followed by a wash with PBS, and incubated at 37 C for 15 min in 40 μl HBSS containing 125U Benzonase and 5 mM magnesium chloride. DNA was extracted using the QIAamp DNA mini kit (Qiagen) following the manufacturer's protocol with the

addition of 1 mM EDTA added to buffer ATL. Samples were then analyzed using qPCR with TaqMan Fast Advanced Master Mix primers and probes using 7500 Real-Time PCR System (Applied Biosystems). The cycling conditions were 50 C for 2 min, then 95 C for 2 min, followed by 40 cycles of 15 s at 95 C and 1 min at 60 C. For *POLQ* CRISPR KO, samples were normalized to the NHEJ substrate whereas *LIG4* CRISPR KO samples were normalized to the TMEJ substrate. Equal amounts of DNA were used for samples treated with inhibitors. The wild-type or untreated samples were used as a reference to calculate ΔΔCt values.

## Apoptosis via cleaved caspase

To assess cell death that might result from the cell treatments above, we monitored cleaved caspase-3, a marker for apoptosis. Cells were treated and prepared as described above for synchronization and flow cytometry with the addition of cleaved caspase-3 antibody (Cell Signaling Technology) conjugated to Alexa Fluor 647 and added together with pH3 antibody. As a positive control for apoptosis, cells were separately treated with 10 μM etoposide (Sigma) for 72 hrs. Data were assessed for the percentage of cells positive for cleaved caspase-3.

## svCapture library preparation and sequencing

At least 200,000 cells were collected and centrifuged for 10 min at 1000 rpm from either bulk asynchronous cultures or flow-sorted cell-cycle phases. Supernatant was removed until 100 μl remained and 200 μl DNA/RNA shield (Zymo Research) was added with 15 μl 20 mg/ml proteinase K and incubated for 20 min at room temperature. Genomic DNA was purified using Quick-DNA microprep plus kit according to the manufacturer's instructions (Qiagen). Further processing steps through high-throughput sequencing were performed at the University of Michigan Advanced Genomics Core. Bead-based tagmentation libraries were prepared with the Illumina DNA Prep with enrichment kit, using 300 ng of genomic DNA, IDT for Illumina unique dual barcodes, and library PCR amplification of nine cycles. Libraries were quantified using Qubit and quality checked using an Agilent TapeStation to ensure that at least 350 ng and preferably 500 ng of prepared library was available to support robust target capture.

Hybridization capture probes were targeted to the central 250 kb or 400 kb of large CFS genes, the region of peak accumulation of SV breakpoints[17]. Final probes were designed to be target-specific and synthesized by Twist Biosciences using their proprietary algorithms and used as provided by the vendor. Capture was performed by pooling 500 ng of each library and hybridizing with 4 μl Twist Biosciences probes and 6 μl PCR grade water. Target enrichment on magnetic beads was performed according to manufacturer instructions. Retained library fragments were amplified with 12 cycles of PCR for sequencing.

Sequencing reads were obtained in the 2 × 150 format using Illumina NovaSeq 6000 or Illumina NovaSeq X Plus. Barcoded samples were pooled and subjected to a sequencing depth calculated to yield a projected coverage of ~2,000-fold in the capture target regions based on experience[36], typically 2.5% of a NovaSeq S4 flow cell per sample. Insert size and target region coverage were maintained over narrow ranges over all analyzed samples (Supplementary Fig. 1B).

## svCapture data analysis

**svCapture pipeline execution.** We previously reported the svCapture data analysis pipeline and Shiny app[36] constructed in the svx-mdi-tools suite of the Michigan Data Interface (MDI). Version 2.0.0 of the tool suite or higher was used for most data analysis, contemporary with version 3.0.0 of the genomex-mdi-tools suite dependency and versions 1.3.2 and 1.8.2 of the mdi-pipeline-frameworks and mdi-apps-framework, respectively. Additional program dependency versions are set by the conda environment definitions tied to tool suite versions.

The following steps match previous pipeline descriptions[36]: (i) read trimming, merging, and quality filtering using fastp[98], (ii) read alignment to the genome using bwa mem[99], (iii) aggregation of reads

into read groups representing unique source DNA molecules, and (iv) SV junction detection using discordant alignment of paired reads and split reads. We applied the 'align', 'collate', and 'extract' pipeline actions to individual samples to discover potential discordant read alignments. The 'find' action was then applied at once to all samples sequenced together in an experimental batch to find candidate SV junctions unique to one sample. The GRCh38/hg38 genome was used for all analyses and capture target regions were padded on each side by 800 kb to set the allowable SV breakpoint regions.

Additions to the svCapture pipeline for this work relate to integrating results across multiple experimental batches, delivered by the 'assemble' pipeline action and svCaptureAssembly Shiny app initialized in svx-mdi-tools v1.8.0. These tools apply standardized SV filtering, coverage assessments, and junction analysis, and assemble SV, target, and sample-level metadata into tables. We created individual assemblies for each cell line and another with all cell lines together. Most figures were generated using the svCaptureAssembly app working from those assemblies.

**SV filtering.** Filtering when counting SVs is essential to ensure that true, on-target SVs are counted in preference to read artifacts. Our goal throughout was to count only de novo SVs that arose during the experimental window prior to their expansion by replication. Accordingly, we only counted SV junctions found in a single source DNA molecule in a single sample as defined by the molecule's outer endpoints. Those source DNA molecules needed to be sequenced by at least three redundant read pairs to support their validity, since chimeric PCR artifacts arising late in PCR typically have only one matching read pair[36]. Additional filters included (i) a required mapping quality of 30 or higher in one flanking alignment and 20 or higher in both, (ii) a requirement that at least one SV breakpoint fell in a capture target, with the other falling in a padded target region as defined above, and (iii) exclusion of deletion and duplication SVs less than 10 kb or greater than 1.2 Mb to match the established properties of SVs arising at CFSs[17]. Inversions were filtered to exclude events less than 50 kb due to a known artifact class in transposase libraries of false small inversions with large microhomology tracts resulting from intramolecular hybridization and synthesis during end-filling of Tn5-cleaved DNA ends (Supplementary Fig. 1C)[36,100].

**Target coverage assessment.** Calculating effective target region coverage is essential for comparing samples. Unlike single-nucleotide variants (SNVs), not all bases in reads are equally able to report on the existence of a true SV. SV junctions cannot be detected near the ends of reads because a minimal span of approximately 20 bases must be properly aligned to the genome on each side of the junction. Accordingly, we adjusted non-SV source DNA molecules to ensure that only bases that could have reported an SV junction were included in coverage calculations. Source molecules with fewer than three read pairs were excluded, and the length of the remaining molecules was adjusted by subtracting 2 x 20 bp = 40 bp from the actual length to disregard terminal bases where SV junctions could not be called. Target region coverage was calculated as the sum of all adjusted, on-target source molecule lengths divided by the summed length of all unpadded target regions. For visualization, base-level coverage was averaged over 100 bp bins. Coverage is not uniform throughout target regions due to variable capture probe efficiency, read mappability, and underlying clonal SVs in cell lines. However, these systematic variations applied similarly to all samples of a cell line processed with the same capture probes.

## SV junction analysis

**SV junction types and local structures.** The svCapture pipeline reports without preference each of the four canonical types of SV junctions – deletions, duplications, inversions. and translocations[1] –

where, as applied here, the first three types arise in a single capture target while translocations join two different targets. Importantly, short reads can only detect junction sequences consistent with end joining; junctions arising through long blocks of homology are invisible to svCapture[1,36]. To characterize junctions, the last aligned bases nearest the junction on either side were defined as the two breakpoint positions in two coordinate systems: the reference genome and the SV-containing source DNA molecule. The distance between the junction positions in the reference genome defined the SV size. The junction was labeled as a microhomology event if the two junction positions overlapped in the source molecule such that the same read bases aligned to both reference breakpoints. The breakpoint positions of blunt joints abutted in the source molecule, whereas de novo insertions were evident as read bases between the breakpoint positions that did not align to either reference breakpoint. We plot insertions as positive offsets of breakpoint positions in the SV molecule and microhomologies as negative offsets, i.e., overlapping alignments.

To compare junction profiles between experimental groups, we first calculated the average microhomology length of all junctions that did not have a de novo insertion, e.g., R expression 'mean(microhomologyLength[microhomologyLength >= 0])', to reveal the strand annealing preferences of the underlying mechanism(s). We further calculated the fraction of all junctions that had a de novo insertion between 2 and 15 bp, inclusive, to reveal the extent to which those mechanisms supported potentially templated insertions.

**Insertion template discovery and characterization.** Some novel bases inserted at junctions are known to be copied from template bases near the reference genome breakpoints[46,47,66]. Locating templates is complicated by the fact that inserted sequences are often too short to be uniquely identified. However, base constraints when comparing a junction to a candidate template extend beyond the inserted bases to include priming and resolving microhomologies used during repair[46]. To maximize sensitivity and specificity while searching for possible insertion templates, we adjusted the required number of flanking microhomology bases to ensure that a total of at least seven bases with one base of flanking homology on each side were used as a template query sequence. Thus, a single-base insertion was queried using three bases of initial microhomology on each side, a four-base insertion was queried using two bases of microhomology on each side, and insertions of five or more bases were queried with one base of microhomology on each side.

We searched for exact matches of the minimal search sequence for a given insertion junction in both strands of both reference genome breakpoints in a span 500 bp upstream and downstream of the junction position. Thus, we searched widely over the reference bases that were both retained and lost at the junction over a total of 2 breakpoints x 2 strands x 2 junction sides x 500 bp per side = 4 kb genomic sequence. When a template match was found, the flanking microhomologies were expanded from the initial query to the maximum possible extent, stopping just before the first mismatch between the junction and template sequences. If multiple possible matches were found, we preferred the template with the longest span, including the flanking microhomologies, or, if that did not differentiate them, the template closest to the junction.

The type of the final selected insertion template, if any, was defined by its strand and placement in the reference breakpoints. As shown in figures, relative to a deletion SV junction, foldback and palindrome insertion templates were found in retained genomic sequence on the bottom genome strand, cross-junction templates were found in retained genomic sequence on the top strand, strand-switching templates were found on the bottom strand at least partially in the lost sequence, and expansion templates were found on the top genome strand in a manner that crossed the breakpoint junction position from retained into lost sequence. Importantly, retained and lost segments are defined relative to breakpoint positions, as defined above, not to the position of the underlying DSB ends, which cannot be ascertained from junction sequences.

## Quantification and statistical analysis

**Number and source of experimental replicates.** Throughout, figures are labeled to indicate the number of SVs or other relevant input counts that contributed to each plot. The number of samples contributing to each experimental group is evident from the plotted sample data points and enumerated in figure legends. Because of the time and expense involved in svCapture experiments, it was not possible to repeat all controls in all experimental batches. Accordingly, we plot all relevant data points that match an experimental group regardless of when they were acquired and indicate the shared experimental batches of different data points by their sample point color.

In general, our approach was to analyze sufficient replicates until the relationship between key experimental groups was established by statistical methods below. However, for completeness in reporting results, we sometimes show supplemental experiments performed in a single replicate with a given cell line. In such cases the data relationships match results from other cell lines and do not form the primary basis of data interpretation; single-replicate experiments should always be interpreted with caution and integrated in this way. Error bars on SV frequency plots represent the mean +/− 2 standard deviations or two or more data points.

**Comparing intergroup SV Frequency.** svCapture is a Poisson process in which an integer number of SVs of a particular class, most notably de novo deletions, are detected over an interval manifest as the library read depth, i.e., more SVs are detected the more deeply a sample is sequenced. Thus, the estimated Poisson rate for an svCapture sample is the number of detected SVs divided by the on-target coverage, noting that target regions were the same size in all samples from a given cell line. We label this Poisson rate SV Frequency to emphasize the intuitive sense that it approximates the fraction of aggregated target alleles in cells, i.e., target haploid equivalents, that carried a de novo SV. Consistent with prior microarrays results[17] this fraction can exceed 50% of alleles under replication stress, reflecting the high mutational potential of the CFS hotspot genes we targeted.

Inter-sample differences in factors such as APH preparation potency, library insert sizes, fold target enrichment, and complexity are expected to influence SV count variance beyond random sampling, so we modeled svCapture data as an overdispersed Poisson using the negative binomial distribution (NBD). Moreover, svCapture results aggregate libraries prepared and sequenced over several years. Although data show a high degree of reproducibility, batch effects such as unidentified differences in conditions and kits over time might also contribute to inter-sample variance.

To appropriately compare SV Frequency between experimental groups, e.g., wild type vs. mutant, we performed pairwise intergroup comparisons using a generalized linear model based on the NBD in which the number of detected SVs, nSVs, varied as a function of the experimental group and batch as independent covariates. Target region coverage per sample was included as an offset parameter of slope 1 to effectively model SV Frequency. The relevant R language expression is 'fit = MASS::glm.nb(nSvs ~ group + batch + offset(log(coverage)))', where 'group' and 'batch' are categorical variables and the $p$-value of the intergroup comparison was obtained from the 'group' variable as 'coef(summary(fit))[2,4]'. When glm.nb failed to return a result, the $p$-value was calculated using the glm function poisson model without overdispersion; such cases are denoted with a dashed significance line in plots. Throughout, we used $p <= 0.01$ as a significance threshold, with plot labels *, $p <= 0.01$; **, $p <= 0.001$; and ***, $p <= 0.0001$.

**Assessing insertion template enrichment.** Some insertion templates are expected to be found locally by chance at a Poisson rate of mu = 4 kb search space x $1 / 4^{nTemplateBases}$. The probability of finding at least one local match corresponds to R expression 'trialSuccessProb = 1 - dpois(0, mu)'. To determine the statistical significance of the actual number of templates found, we took each junction sequence at a given insertion size as an independent Bernoulli trial and estimated the *p*-value using R expression '1 - pbinom(nFound − 1, nSearched, trialSuccessProb)', where nSearched and nFound are the number of searched and found insertion templates, respectively, and the resulting *p*-value is the likelihood of finding nFound or more templates by random chance. A *p*-value < 0.05 was taken as significant evidence for the contribution of local templates to the appearance of insertion junctions.

## Reporting summary

Further information on research design is available in the Nature Portfolio Reporting Summary linked to this article.

## Data availability

svCapture sequencing data have been deposited in two repositories with sample lists provided as Supplementary Data 1. Data from cell line UM-HF1 require human data access restrictions and were deposited into the Database of Genotypes and Phenotypes (dbGaP) and are available with the dbGaP Study Accession phs003121.v2.p1 [https://www.ncbi.nlm.nih.gov/projects/gap/cgi-bin/study.cgi?study_id=phs003121.v2.p1]. Data from commercially available cell lines HCT116 and GM12878 were deposited into the NCBI Sequence Read Archive (SRA) with the BioProject ID PRJNA1085257. Flow cytometry data were deposited into Mendeley Data and are available at https://doi.org/10.17632/mz53d2486n.1[101]. The main processed data outputs of the svCapture pipeline are provided as Supplementary Data 2 and 3, included in the Zenodo code set linked to GitHub alongside the job scripts that generated them (see below), or in a separate Zenodo dataset carrying larger output files at https://doi.org/10.5281/zenodo.10916986[102], including data packages and app bookmarks. Source data are provided with this paper.

## Code availability

Code comprising the svCapture data analysis pipeline and app can be found at GitHub (https://github.com/wilsontelab/svx-mdi-tools/tree/v2.0.3)[103]. Data-specific job scripts used to execute the pipeline for samples in this manuscript and associated support files, including resource files, sample lists, and job logs, can be found at GitHub (https://github.com/wilsontelab/publications/tree/main/CFS-M_phase-PolQ-2023)[104].

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

## Acknowledgements

This work was funded by grants CA200731 and GM147026 from the National Institutes of Health to T.E.W. and T.W.G. We thank Dale Ramsden for reagents, advice in establishing TMEJ/NHEJ assays, and for valuable discussion. We thank Pamela Bennett-Baker for assistance with reagent preparation and method validation in early stages of the work and Charles Kazazian for assistance with cytogenetics. We thank the University of Michigan Advanced Genomics Core for skilled handling of svCapture library preparation and sequencing, and the University of Michigan Flow Cytometry Core for expert assistance with long flow sorting sessions. We thank Patrick O'Brien and Martin Arlt for critical reading of the manuscript and valuable input over many years.

## Author contributions

T.E.W. designed the experiments, wrote the data analysis pipeline, analyzed data, and wrote the manuscript; S.A. designed and performed the experiments and analyzed data; A.W. assisted in the design and performance of experiments; T.W.G. designed the experiments, performed cytogenetic analysis, analyzed data, and wrote the manuscript.

## Competing interests

The authors declare no competing interests.
