## [Transparent Peer Review file · Nature Communications]

Replication stress induces POLQ-mediated structural variant formation throughout common fragile sites after entry into mitosis

Corresponding Author: Dr Thomas Wilson

Version 0:

Reviewer comments:

Reviewer #1

(Remarks to the Author)

In this study, Wilson et al further advance their work and the field on the generative mechanism of structural variations that occur at common fragile sites. They employ their recently developed svCapture technology to determine the configuration of de novo SVs at nucleotide resolution. After having established that SVs induced by applying replication stress are predominantly formed at mitosis, they (to me) convincingly rule out MiDAS of being involved, but instead establishes TMEJ as the underlying mechanism, first by a detailed analysis of the junctions of thousands of de novo deletions, then by genetic dissection (using POLQ knockout and inhibitors).

I found the manuscript of outstanding caliber on all aspects: technical finesse, elegant design of experiments, appropriate controls, depth of the analysis, readability, etc. Uncommon to my critical nature I have nothing to comment on, or have concerns about the study and the conclusions. I feel it is truly impressive and convincing work.

Just a few items the authors may wish to think about, but these are not critical:

*) The study of the templated insertions is already strong and convincing. However, by using a search space of 500 bp up and downstream, while finding later that almost all mappable insertions are within 20 bp from the junctions (e.g. Fig 4F and S7a), one could also limit the search space to e.g. 50 bp, which will increase the statistics, and allowing for shorter insertions to be reliably mapped.

*) The authors provide a speculative model for the insertions that map across the deletion junctions, which I consider unlikely. While I'm perfectly fine with letting it in as it is (it is, above all, speculation), I wondered whether the authors considered the option that "another molecule" (i.e. the sister chromatid) has served as a template. It is still debatable whether (all) inverted insertions are the product of intramolecular foldback or, instead, the product from one DSB end performing a (abrogated) TMEJ reaction with a broken sister chromatid having the DSB in close proximity (see e.g. Fig. 3 iv of ref. 46). Consider the option (as drawn out in Figure 4I) that both sisters have DSB ends. Because these ends are not necessarily ending at the identical nucleotides, a shorter end (on one side) could use a longer end (on the same side of the broken sister) as a template hence resulting in a TMEJ insertions that map across the deletion junction. This explanation is most parsimonious as it doesn't require any other (new) biology.

*) Figure 4D was not very intuitive to me and I'm still not sure I'm understanding it.

*) The discussion is a bit lengthy (it is a very clearly written manuscript and the biology is not too complex, hence I felt that not all results needed to be reiterated in the final section).

As said, the above comments are just considerations and I want to compliment the authors for a magnificent piece of work.

Reviewer #2

(Remarks to the Author)

In this manuscript, Wilson et al. analyzed genomic structural variants (SVs) formed at specific common fragile sites (CFSs) under replication stress. From several experiments, the authors reached the conclusion that mitotic DNA synthesis (MiDAS)

does not participate in SV formation, but DNA polymerase theta (POLQ)-mediated end-joining (TMEJ) is responsible for SV formation, primarily in M phase of the cell cycle. Although these findings are potentially interesting, the finding that POLQ is important in mitosis is not surprising (cf. Brambati A. et al. Science, 2023; Gelot C. et al. Nature, 2023). I strongly recommend the authors to perform additional experiments to solidify their results and reorganize the main text thoroughly to make the paper more attractive. Besides the writing style of the manuscript, another problem is that most of the figures are difficult to understand and the figure legends do not provide sufficient detail of information, making it difficult to follow the manuscript. Specific comments follow:

1. The authors only used aphidicolin to induce 'replication stress'. Some other drugs (such as HU and CPT) should be used for the experiments, or the 'replication-stress' in the title should be changed to aphidicolin.
2. The authors state that MiDAS is inhibited by aphidicolin, while the same agent, albeit at a lower concentration, is used to induce 'replication stress'. This may not be a good experimental design because if you inhibit MiDAS, then more 'replication stress' is induced.
3. The main impact of this paper is POLQ involvement in SV formation. Nevertheless, the evidence is not solid as the authors presented different results from genetic deletion and chemical inhibition, with the latter different outcomes between the three cell lines. Multiple POLQ knockout cell lines should be used to establish the POLQ involvement.

Reviewer #3

(Remarks to the Author)

In this manuscript, Wilson and colleagues establish a novel role for POLQ in the formation of structural variants induced by replication stress at common fragile sites. The findings are novel and will be of general interest for the readers of Nature Communications. I have only few comments, as the manuscript is already strong and the data compelling.

The authors first establish their experimental system, using svCapture at common fragile sites after induction of replication stress. This figure and the underlying data are very difficult to grasp, and better graphical visualization of the data is necessary to improve the readability of the paper. For instance, the concept of "negative insertion" is counterintuitive, to say the least. The SV size should also be numbered in raw numbers, not log 10 bp. Finally, the svCapture techniques should be better explained – in the figure as well as the text.

The authors then go on to characterizing the cell cycle stage at which the SVs occur and establish that SVs accumulate in M rather than S or G2. The analysis of both drug-synchronized cells and asynchronous-cell sorted cells is very compelling. What is missing in this figure is the number of SVs in G1, especially considering the final data with POLQ KO. Authors should analyze SVs in cells released into G1 after RO3306 wash.

They then show that SV junctions are frequently associated with microhomologies reminiscent of the TMEJ pathway. Again, Figure 4 is quite difficult to read. For instance, figures S6A-B should be re-integrated in the main figure for better compression, rather than panels 4H-I. 4B is also difficult to comprehend.

Figure 5 then defines POLQ as an essential contributor of SV formation during mitosis, while Figure 6 establishes that NHEJ is responsible for POLQ-independent SV formation. I have two comments here. First, it is unclear whether NHEJ is normally used for SVs or if it is a backup when POLQ is absent. Characterization of the occurrence of SV deletions during G1 vs M in WT vs. POLQ KO cells would answer that essential question. Secondly, the authors use indifferently POLQ and TMEJ as terminology. Though they have shown that POLQ is involved, it could be through a mechanism slightly different than the classic TMEJ pathway. Ideally, other factors of TMEJ should be tested. Otherwise, this point should be discussed.

In general, the findings that POLQ plays a major role in the formation of SVs pose the question of SV formation in HR-deficient cancers, which heavily rely on TMEJ and which are likely to struggle more with replication stress. The authors could integrate that point in the discussion.

Minor comment: ART558 should be removed from Figure 3A.

Version 1:

Reviewer comments:

Reviewer #1

(Remarks to the Author)

I already was impressed by this work and felt that the conclusions drawn from the data were sound. To me, the authors further improved the readability of their manuscript.

I thus fully support publication of this study.

Reviewer #2

(Remarks to the Author)

The revised manuscript is much improved. However, to make this paper more understandable to a broader range of scientists, more improvements and additional data are needed. Many of the graphs wrongly have error bars for only one or two points (Figs. 1E, 2DFK, 3BCD, 5CDEF, 6ABC, S1EFG, S4C, S9, and S10ABC). How was statistical analysis performed for these data? Some of these may be acceptable (e.g., Figs. 1-3, 5, S1, S4), but additional experiments should be performed for the others, especially Figs. 5 and 6. This is particularly important because the main conclusion of this paper, POLQ's involvement in SV formation, is not solid. It is evident that TMEJ does not explain all SVs as the authors admit, so even the title is misleading. Specific comments follow:

1. The authors should clarify where the nicks arose on each DNA strand. Otherwise, it should be stated that the edge of the deletion site is assumed to reflect the break site and the actual position of the break is uncertain. This is important and needs to be much more clearly specified in the manuscript and in the figures and legends.

2. Please explain why the points in Fig. 1C are not whole numbers. Is the length of microhomology inferred? If so, perhaps the authors could explain the reasoning behind their algorithm.

3. Aphidicolin inhibits all replicative DNA polymerases (alpha, delta, epsilon) (and do not inhibit DNA repair polymerases?). This fact must be clearly stated in the Introduction.

4. Please explain what is drawn from Fig. S9 results. Why does ART558 have an effect in TMEJ-deficient cells? Also, recent work suggested the involvement of pol delta in TMEJ. The authors may want to check if aphidicolin affects TMEJ using the Fig. S9 assay method.

5. The Abstract should be toned down as the authors overextend their conclusions. For example, 'as it occurred in cells' should be cautiously be phrased as 'as it occurred in human cell lines'.

6. In the first sentence of the third paragraph of the Introduction, the authors state that "... an experimental model for non-recurrent CNV formation". However, this is not applicable to all non-recurrent CNV sites and therefore needs to be rephrased. Also, in the last sentence (L99), "in normal and cancer cells" should be removed.

7. The authors need to point out the limitations of their study; otherwise, the misleading aspects can have a negative effect in this field. For example, creating so many aphidicolin breaks may not be physiologic. Also, there is no evidence that supports that their findings are true for the entire cell cycle and for normal cells.

--

Reviewer #3

(Remarks to the Author)

The revised manuscript from Wilson and colleagues is improved in clarity, and the points that I asked to be discussed are now included. The main experiment I was asking for was a G1 analysis of SVs. I do understand however the arguments made by the authors that this is a massive undertaking, and somehow out of the scope of the manuscript. The current analysis and findings during mitosis are, per se, sufficient to warrant publication. In conclusion, I believe that the modifications made by the authors are satisfactory and I support the publication of this revised manuscript in Nature Communications.

Version 2:

Reviewer comments:

Reviewer #4

(Remarks to the Author)

The authors provide a timely, definitive identification of M-phase activity of Pol θ as a major determinant of replication stress-induced SV's at CFS. The following suggested changes could be considered.

1) The authors are appropriately inclusive of "singlicate" data, as these less stringent examples of their data nevertheless help argue for consistency across e.g. cell line models. In some cases (e.g. Fig 1E), this data might be better moved to the supplement. Additionally, per journal format it would be helpful to include in the figure legend a little more of the methods section detail regarding the applied statistical analyses; the current comment "...with selected Poisson-based intergroup p-values..." is a little vague.

2) The authors should probably add a comment in the introduction regarding past work arguing the extent low-dose aphidicolin is a generalizable model for induction of replication stress and SV formation at CFS. Additionally, it may be worth expanding in the discussion as to the importance of DSBs as the likely Pol θ engaging intermediate when considering their model (current mention of MUS81 and GEN1) vs. engagement of Pol θ in repair of gaps using a different source of replication stress (BRCA deficiency; see e.g. Belan et al, Mol. Cell). Additional speculation as to the possibility that different types of replication stress, and the different types of damage they generate (gaps, one ended breaks, and two ended breaks), help determine the repair pathways engaged is perhaps worth mentioning.

REVIEWER COMMENTS

Reviewer comments are in black, our responses are in blue.

General comments, multiple reviewers

We thank the reviewers for the many positive comments and helpful suggestions. Some reviewers suggested substantial amounts of new data. We agree that more data are generally better, including the goal of G1 analysis addressed in response to Reviewer 3, but it is important to understand what generating svCapture data entails. In many figures, entire high-throughput sequencing runs are reduced to one data point on a plot. Generating an entire new series of such plots with replicates is a large, expensive, and time-consuming task. It took many years and many tens of thousands of sequencing dollars to accumulate the current data. Large new data extensions are beyond the scope of this study.

Fortunately, we already provide a unique and previously missing data class that demonstrates and extends important concepts about SV junction formation at replication-stress-induced DSBs in chromosome fragile sites and in mitosis. The value of characterizing >48K SV junctions acquired in controlled, prospective experiments is clear in our study, which thoroughly addresses its target points with current data. We directed these first efforts with svCapture at critical outstanding questions and factors nominated in prior studies for which prospective genetic SV data were an essential unmet need.

In addition, we are pleased to add a last new panel in revised Figure 6, addressing SV junction formation in S-phase, which reinforces how NHEJ acts in backup to TMEJ. Adding these data was a modest extension of current studies, in contrast to G1, which would require nearly as much new data as we already provide.

Given these and the reviewers' comments, we worked throughout the text to ensure that we are not overstating the scope of our work and conclusions to remain on point about the value and meaning of the extensive data we provide on POLQ and S, G2, and M-phases.

Reviewer #1:

In this study, Wilson et al further advance their work and the field on the generative mechanism of structural variations that occur at common fragile sites. They employ their recently developed svCapture technology to determine the configuration of de novo SVs at nucleotide resolution. After having established that SVs induced by applying replication stress are predominantly formed at mitosis, they (to me) convincingly rule out MiDAS of being involved, but instead establishes TMEJ as the underlying mechanism, first by a detailed analysis of the junctions of thousands of de novo deletions, then by genetic dissection (using POLQ knockout and inhibitors).

I found the manuscript of outstanding caliber on all aspects: technical finesse, elegant design of experiments, appropriate controls, depth of the analysis, readability, etc.

Uncommon to my critical nature I have nothing to comment on, or have concerns about the study and the conclusions. I feel it is truly impressive and convincing work.

Just a few items the authors may wish to think about, but these are not critical:

*) The study of the templated insertions is already strong and convincing. However, by using a search space of 500 bp up and downstream, while finding later that almost all mappable insertions are within 20 bp from the junctions (e.g. Fig 4F and S7a), one could also limit the search space to e.g. 50 bp, which will increase the statistics, and allowing for shorter insertions to be reliably mapped.

We appreciate the reviewer's highly positive assessment of our manuscript. The suggested revised search space would only change the nature of the reported values a small amount and no new templates would be found or be placed differently. The biggest change is that a fraction of templates found within 500 bp would be lost with a narrower window. Our search process could find multiple hits for any query sequence and when multiple equivalent hits were found, we favored placement closest to the DSB. Thus, we would not find more templates than we did. We do not favor relaxing search stringency, e.g., by allowing shorter templates, as one quickly runs into the problem that templates are found by random chance too frequently to be interpreted. This reflects the biology – TMEJ exploits short sequences encountered by the enzymes in ~any sequence context.

*) The authors provide a speculative model for the insertions that map across the deletion junctions, which I consider unlikely. While I'm perfectly fine with letting it in as it is (it is, above all, speculation), I wondered whether the authors considered the option that "another molecule" (i.e. the sister chromatid) has served as a template. It is still debatable whether (all) inverted insertions are the product of intramolecular foldback or, instead, the product from one DSB end performing a (abrogated) TMEJ reaction with a broken sister chromatid having the DSB in close proximity (see e.g Fig. 3 iv of ref. 46). Consider the option (as drawn out in Figure 4I) that both sisters have DSB ends. Because these ends are not necessarily ending at the identical nucleotides, a shorter end (on one side) could use a longer end (on the same side of the broken sister) as a template hence resulting in a TMEJ insertions that map across the deletion junction. This explanation is most parsimonious as it doesn't require any other (new) biology.

We thank the reviewer for this suggestion and agree that use of a sister chromatid as a repair template is possible, which is partly why we thought it important to use a wide template search space. We added new mechanism diagrams to Figure S6 and S7 to illustrate this possibility for both inversion/foldback and the novel insertion class. We adjusted associated text in Results and Discussion to clarify that there are other possible sources of these templates and that our data cannot declare the mechanistic sequence, only the outcome. These additions brought clarity on several points. First, sister-chromatid synthesis is a reasonable explanation for the observed apparent palindromic inverted insertions. However, we also now emphasize more strongly that the shift in foldback/inverse templates away from DSB ends provides strong evidence that most inverted insertions do truly arise from unimolecular foldback synthesis. It is worth noting

that we while we could detect templates in the inversion orientation that sampled the sister chromatid in advance of the junction position, few in any were observed.

Importantly, the novel insertion class may only appear to have templates that cross the junction breakpoint. The goal of any model for these events is to describe how that situation can arise. Our models recognize that these insertion events occurred at short direct repeats where strand slippage is a reasonable inference. Following repeat expansion, the algorithm is fooled into thinking the template crossed the apparent breakpoint. We had not previously made clear that the novel class occurs via repeat expansion, which is now corrected, including renaming this event class as “expansions”, which is descriptive, rather than “slippage” which presupposes a mechanism.

We do not think the slippage model for expansions invokes new biology and favor it to because we find it to be the most parsimonious in that it is unimolecular. The net insertion in the slippage model requires only one DSB end, end-filling, and strand slippage, all of which are well described. The sister-chromatid-dependent alternative requires a bimolecular reaction (trimolecular if you include the ultimate SV partner DSB), strand displacement, and unwinding. Regardless, we have rewritten these sections to emphasize that more work is needed to explore fork cleavage polarity leading to SV formation.

*) Figure 4D was not very intuitive to me and I'm still not sure I'm understanding it.

Figure 4D plots the yield of templates found in the search space as a function of insertion size, thus showing that we found fewer templates as insertion size increased. The blue line is the yield expected by random chance, as labeled on the plot and stated in the legend. We clarified the axis, changing “% Found Templates” to “% of Templates Identified”. The most confusing part of Figure 4D was the numbers on the random chance line denoting the number of bases of flanking microhomology demanded in the search, which was higher at smaller insertion sizes to ensure that at least 7 bp were present in the query. Given the reviewer's comment, we opted to remove those numbers from the figure and state in the legend and methods how that line was calculated and what template sizes were required.

*) The discussion is a bit lengthy (it is a very clearly written manuscript and the biology is not too complex, hence I felt that not all results needed to reiterated in the final section).

We agree and have worked to substantially shorten the Discussion by reducing the extent of repetition of Results and other changes.

As said, the above comments are just considerations and I want to compliment the authors for a magnificent piece of work.

Reviewer #2:

In this manuscript, Wilson et al. analyzed genomic structural variants (SVs) formed at specific common fragile sites (CFSs) under replication stress. From several experiments,

the authors reached the conclusion that mitotic DNA synthesis (MiDAS) does not participate in SV formation, but DNA polymerase theta (POLQ)-mediated end-joining (TMEJ) is responsible for SV formation, primarily in M phase of the cell cycle. Although these findings are potentially interesting, the finding that POLQ is important in mitosis is not surprising (cf. Brambati A. et al. Science, 2023; Gelot C. et al. Nature, 2023). I strongly recommend the authors to perform additional experiments to solidify their results and reorganize the main text thoroughly to make the paper more attractive. Besides the writing style of the manuscript, another problem is that most of the figures are difficult to understand and the figure legends do not provide sufficient detail of information, making it difficult to follow the manuscript. Specific comments follow:

Regarding additional experiments, please see comments at the top of our response. Our paper synthesizes large amounts of hard-won sequencing data we think are unique in the literature and needed for making conclusions about things suggested in works noted by the reviewer. POLQ roles in mitosis have indeed become apparent in many studies, but none presented the essential data class and associated findings we now do. Our work strongly complements and does not reproduce those prior works.

Regarding presentation, we have worked hard to communicate a complex body of technical work with a new method in an understandable way. All plot axes and other figure elements are clearly labeled, and all figure panels are explicitly described in order in Results. In the absence of specific suggestions, we worked in the revision to label multiple figures more clearly, added details to figure legends, and worked to be as clear as possible. We draw the reviewer's attention to improved labeling in Figure 1C and 1F and others with similar microhomology/insertion and SV size axes, and in Figure 4 and Figure S6/S7.

1. The authors only used aphidicolin to induce 'replication stress'. Some other drugs (such as HU and CPT) should be used for the experiments, or the 'replication-stress' in the title should be changed to aphidicolin.

Reducing the title to the chemical name of a specific reagent could unnecessarily confuse potential interested readers. Low dose aphidicolin is a well-established and the most used model of replication stress across many experiments and papers. It is the definitive reagent used to induce replication stress at common fragile sites, which the title identifies as the scope of the study. Instead, we revised the Abstract sentence that summarizes our experimental approach to state that we specifically used aphidicolin. Aspects of the revised Discussion also address the importance of studying alternative forms of replication stress in future works, especially in cancer.

2. The authors state that MiDAS is inhibited by aphidicolin, while the same agent, albeit at a lower concentration, is used to induce 'replication stress'. This may not be a good experimental design because if you inhibit MiDAS, then more 'replication stress' is induced.

These distinct uses of aphidicolin (as well as other agents in other studies such as hydroxyurea) as a function of dose are well established, most notably by the Ian Hickson group in his seminal studies of MiDAS. Low-dose APH (or other replication inhibitors) is

well known to create replication stress by impairing the progression of replication forks in S-phase. Forks fail at an increased rate, but as we emphasize in this and our papers over many decades, replication does proceed. High-dose APH was restricted to M-phase as we used it, so cannot induce replication stress in the same manner as it was never present in S-phase in cells subjected to sequencing. However, as shown first by Hickson and verified in our manuscript, high-dose APH restricted to M-phase abrogates MiDAS without impacting the progression of cells through mitosis.

Our goal was to establish whether MiDAS could be the mechanism by which SV junctions formed. We show both that we effectively suppressed MiDAS as measured by mitotic EdU incorporation and that SV junctions still formed, so definitive conclusion could be made that MiDAS was not the mechanism we sought, despite being active in M-phase.

3. The main impact of this paper is POLQ involvement in SV formation. Nevertheless, the evidence is not solid as the authors presented different results from genetic deletion and chemical inhibition, with the latter different outcomes between the three cell lines. Multiple POLQ knockout cell lines should be used to establish the POLQ involvement.

We report data from two independent POLQ knockout clones in each of two cell lines, one cancer, one “normal”. We also use chemical inhibition with two agents, one more (ART558) and one less (novobiocin) specific and address in Results and Discussion the differences between these paradigms and their potential influence on our results. Results are internally consistent and reproducible within each experimental system. Doing it more times would not change that as experiments are well replicated (unless otherwise noted for some Supplemental Figures) and trustworthy.

The POLQ phenotype is not simple and clear because DSB repair is redundant, as addressed throughout the paper. Moreover, the degree and nature of the action of redundant pathways can depend on cellular context. We discuss how our results reflect the differential extents to which NHEJ (and possibly other mechanisms) might rescue SV junction formation when TMEJ is impaired in different contexts, and why chemical inhibition need not be equivalent to genetic knockout, an observation with important implications. Fortunately, one combination – POLQ KO in HCT116 cells –conclusively shows SV junction loss in M phase to establish our most important conclusion.

Another important factor is the cell cycle. We were not satisfied with our analysis of SV formation in S vs. M phase. While the largest fraction of SVs was clearly induced as cells passed into M, there was a significant increase of SVs in S. We therefore added a new data panel to Figure 6 to address how much NHEJ contributes to SV formation in S phase as compared to M. The new data affirm that POLQ is the dominant pathway with NHEJ having a redundant and synergistic role.

Reviewer #3:

In this manuscript, Wilson and colleagues establish a novel role for POLQ in the formation of structural variants induced by replication stress at common fragile sites. The findings

are novel and will be of general interest for the readers of Nature Communications. I have only few comments, as the manuscript is already strong and the data compelling.

The authors first establish their experimental system, using svCapture at common fragile sites after induction of replication stress. This figure and the underlying data are very difficult to grasp, and better graphical visualization of the data is necessary to improve the readability of the paper. For instance, the concept of “negative insertion” is counterintuitive, to say the least. The SV size should also be numbered in raw numbers, not log 10 bp. Finally, the svCapture techniques should be better explained – in the figure as well as the text.

We thank the reviewer for their positive comments. Some plots are expected to be new to readers because few papers have presented the type of data we collected. Others that have also used plot axes that represent the range of inserted, blunt, and microhomology bases at individual junctions. This is not a contrivance or a convenience. It follows directly from the way junction sequences are analyzed, where overlap, blunt fusion, and novel inserted bases are part of a single continuous spectrum of breakpoint relationships, *i.e.*, offsets (see Methods). Presenting data on a single axis is accurate and essential.

That said, we agree that “negative insertion” was obtuse, so we adjusted axes of this type to use an intuitive label that simply identifies which portion of the plot represents microhomology, blunt, and insertion bases at junctions. All lengths are now indicated as positive numbers while retaining the essential concept of plotting them on a single axis.

We followed the reviewer’s labeling suggestion to use numbers such as 10^4 on plot axes rather than just the exponent 4.

The authors then go on to characterizing the cell cycle stage at which the SVs occur and establish that SVs accumulate in M rather than S or G2. The analysis of both drug-synchronized cells and asynchronous-cell sorted cells is very compelling. What is missing in this figure is the number of SVs in G1, especially considering the final data with POLQ KO. Authors should analyze SVs in cells released into G1 after RO3306 wash.

Regarding additional experiments, please see comments at the top of our response. We agree that studying cells that pass from M phase into G1 is important (it is an aim of our grant supporting this work). However, that is a very large study by itself, not a small addition to this paper. To do it well and correctly, we need to perform experiments in multiple conditions of knockout and chemical inhibition, in multiple replicates, at multiple times after release, and likely with multiple ways of arresting cells in the following G1. We need to correlate outcomes to new cytological markers such as anaphase bridges and 53BP1 foci. Importantly, because we establish that cells accumulate SVs as they pass through M-phase, any new SV signal arising in G1 because of replication stress in S would accumulate on top of an already strong M-phase signal. G1 studies would also be confounded by lower cell purity than we obtain in M, since there are always some cells with 2N DNA content. Finally, we might anticipate G1 SVs to form by a distinct mechanism. These factors require even greater data depth for a G1 study and place new demands on

our analysis pipeline. We hope these comments relate the scope of what is being requested and that it exceeds what we can do in this study.

Since we cannot add this large amount of new data, we worked carefully to ensure that we do not overstate the scope of our study or conclusions. Even without new data we can make important new conclusions about SV formation in M phase that will remain true even as future efforts address the non-exclusive possibility of SV formation in the subsequent G1. We already hit this point in Discussion but now relate the scope and limitations of our study earlier and more accurately. Examples of recurring text changes include favoring “many SVs” over “most SVs form in M” and avoiding phrases like “form preferentially in M”, which were previously overstated the absence of correlative G1 data.

They then show that SV junctions are frequently associated with microhomologies reminiscent of the TMEJ pathway. Again, Figure 4 is quite difficult to read. For instance, figures S6A-B should be re-integrated in the main figure for better compression, rather than panels 4H-I. 4B is also difficult to comprehend.

We made several clarifying changes in Figure 4 (see comments to other reviewers also). Figures S6A-B show foldback and cross-junction template mechanisms that are well described by others, so we did not favor that they be in the main figure. Conversely, Figure 4H was one proposed mechanism for a novel junction class observed in this study. However, because of points raised by Reviewer 1, we needed to expand these models so they are now too big for the main text and are now found in new Figure S7. New Figure 4H is now a depiction of the newly renamed repeat expansion class of insertions. Figure 4I has been made more generic and is a small, simple, but we think helpful, orienting diagram.

Figure 5 then defines POLQ as an essential contributor of SV formation during mitosis, while Figure 6 establishes that NHEJ is responsible for POLQ-independent SV formation. I have two comments here. First, it is unclear whether NHEJ is normally used for SVs or if it is a backup when POLQ is absent. Characterization of the occurrence of SV deletions during G1 vs M in WT vs. POLQ KO cells would answer that essential question.

The question of whether NHEJ is normally used for SV formation or if it is a backup pathway is high on our minds and we agree more data would help in addressing this point. Because the roles of NHEJ are cell cycle dependent, we have added a new data panel in Figure 6 using combinations of NHEJ and TMEJ knockout and inhibition in S phase cells. Our collective interpretation supported by all data is that in no case does NHEJ appear to be a dominant mechanism of SV junction formation (consistent with our prior microarray studies) as little or no change in SV frequency or structure is induced by NHEJ loss alone. The impacts of TMEJ loss are more variable but often significant on their own, with SV formation sometimes reducing further when NHEJ is impaired on top of TMEJ, including in the new S-phase samples.

The question about G1 is complex and addressed above. It is possible that NHEJ could prove to have a more exclusive role in creating some SV junctions in G1 but establishing that point will be a difficult and large new study, as now stated in Discussion.

Secondly, the authors use indifferently POLQ and TMEJ as terminology. Though they have shown that POLQ is involved, it could be through a mechanism slightly different than the classic TMEJ pathway. Ideally, other factors of TMEJ should be tested. Otherwise, this point should be discussed.

This point is well taken, and we revised the text to be clear that POLQ could possibly be acting in one of its other roles, but that TMEJ is by far most likely since we are studying distant SV junctions mediated by short microhomologies with observed outcomes that parallel CRISPR/Cas9 and related DSB-driven studies. Roles of POLQ in things such as gap filling are very interesting and reflect its pleiotropic activities, but it is more difficult to relate them to SV junction formation that is wholly consistent with end-joining.

As noted above, testing a collection of other TMEJ mutants is undoubtedly of interest but beyond our scope given the complexity and expense of the methods underlying our data.

In general, the findings that POLQ plays a major role in the formation of SVs pose the question of SV formation in HR-deficient cancers, which heavily rely on TMEJ and which are likely to struggle more with replication stress. The authors could integrate that point in the discussion.

We agree and have expanded Discussion text considering the unique vulnerability that HR-deficient cancer will have with genome instability when they depend on POLQ. One example we already discussed relates to predictions of BRCA2-deficient cells.

Minor comment: ART558 should be removed form Figure 3A.

Thank you for catching this error, it has been removed.

REVIEWER COMMENTS

Reviewer comments are in black, our responses are in blue.

In the 2nd revision manuscript:

- first revision changes in the manuscript are marked in red text
- second revision changes are marked in blue text

Reviewer #1:

I already was impressed by this work and felt that the conclusions drawn from the data were sound. To me, the authors further improved the readability of their manuscript.

I thus fully support publication of this study.

We thank the reviewer for their strong endorsement of our manuscript.

Reviewer #2:

The revised manuscript is much improved. However, to make this paper more understandable to a broader range of scientists, more improvements and additional data are needed. Many of the graphs wrongly have error bars for only one or two points (Figs. 1E, 2DFK, 3BCD, 5CDEF, 6ABC, S1EFG, S4C, S9, and S10ABC). How was statistical analysis performed for these data? Some of these may be acceptable (e.g., Figs. 1-3, 5, S1, S4), but additional experiments should be performed for the others, especially Figs. 5 and 6.

We use error bars which are correctly identified in Methods and the revised Figure 1 Legend as +/- two sample standard deviations for two or more data points. It is possible and meaningful to calculate a standard deviation with two or more data points as a representation of the range of the data observed. We do not calculate standard deviations for a single data point as it is mathematically impossible and are careful to point out those situations in text. Consistent with comments below, we could provide a confidence interval for a single data point based on the Poisson distribution but choose not to do so for consistency in presentation – our intent of showing error bars based on standard deviation is to reflect sampling overdispersion as described in detail in Methods.

Perhaps more to the reviewer's point, please see our detailed description in Methods, with one small new addition, of how we assessed significance of inter-sample differences in svCapture experiments, echoed in short form in the revised Figure 1 Legend. These assessments are based on Poisson sampling statistics, which is appropriate for svCapture where the data represent a rate of discrete "success" events (an observed SV) relative to independent trials (number of studied haplotype equivalents). It is possible and meaningful to assess the probability that one or two samples – with their observed SV and read counts - are drawn from the same sampling distribution as another set of samples, i.e., that they have same rate of SV formation per haplotype.

To rationalize the points above, it is important to understand that each data point on our SV Frequency plots is not a single measurement, rather, each point aggregates many independent reads across thousands of fold-coverage of our target regions (one goal of Figure 1C is to help readers understand how data are aggregated in subsequent plots). Thus, it is possible and meaningful to ask if one sample with, *e.g.*, 10 observed SVs over 1,000-fold coverage has an SV rate consistent with another sample with 500 SVs over the same coverage, which it clearly does not. With replicates, our statistical methods further account for the impact of overdispersion on that baseline Poisson sampling error.

This is particularly important because the main conclusion of this paper, PolQ's involvement in SV formation, is not solid. It is evident that TMEJ does not explain all SVs as the authors admit, so even the title is misleading.

We do not agree with this assessment on several fronts. A primary issue appears to be that the reviewer perceives we are claiming an exclusive role for POLQ or M-phase in non-recurrent SV junction formation. We are not, and carefully avoided doing so in our first revision. To address this ongoing concern, we made further text changes to avoid overstating our conclusions. Perhaps most importantly, we changed the title to be less declarative about the role of POLQ while giving more weight to the critical timing trajectory of SV formation.

We think the reviewer puts insufficient weight on our detailed junction analysis, which strongly supports a primary role for POLQ in junction formation. Upon passage of cells into M-phase, the bolus of SV junctions we observed has a joining profile that closely matches expectations for TMEJ and shows corresponding changes when POLQ is not present. Even if the rate of SV formation did not change at all, this junction signature pattern would provide powerful evidence of a primary role of POLQ – a singular value of our sequencing approach.

The word “admit” implies we have a bias to arrive at a specific conclusion. In fact, the outcome of this study did not match our original expectations and refuted our entry hypothesis. We objectively describe our data as revealing a non-exclusive but greater role for POLQ in our experimental model than NHEJ, and that there is a clear bolus of new SV junction formation upon entry into M-phase. All extrapolations about SV formation beyond our model to biology use words like “propose” and “possibility”.

Specific comments follow:

1. The authors should clarify where the nicks arose on each DNA strand. Otherwise, it should be stated that the edge of the deletion site is assumed to reflect the break site and the actual position of the break is uncertain. This is important and needs to be much more clearly specified in the manuscript and in the figures and legends.

We infer double-strand break (DSB) formation, not nicks, at stalled replication forks, consistent with the role of end-joining in SV formation. We already described in detail in Results, Discussion, and Methods that our junction analysis is based on breakpoints in observed SV sequences and that these may differ from the exact location of initiating DSBs.

Importantly, breakpoints are most relevant for inferring joining mechanisms. Bases that are deleted away prior to repair have relatively little impact on the resolution mechanism. Extensive processing of DSBs would make conclusions about hotspot positions less robust, a limitation we address in detail in Discussion. This uncertainty is inherent to any study based on unbiased SV detection (as opposed to the use of site-directed DSBs) a limitation offset by the merits of studying a more physiological form of damage than simple DSBs.

2. Please explain why the points in Fig. 1C are not whole numbers. Is the length of microhomology inferred? If so, perhaps the authors could explain the reasoning behind their algorithm.

As stated in the Figure 1C Legend, we added a small amount of random noise to the X axis values (microhomology/insertion size) as a plotting device so that all data points could be visualized. To help prevent others from missing that explanation, we expanded it in the figure legend and added the words “plus random noise” to Figure 1C and Figure S1C X-axis labels.

3. Aphidicolin inhibits all replicative DNA polymerases (alpha, delta, epsilon) (and do not inhibit DNA repair polymerases?). This fact must be clearly stated in the Introduction.

This requested additional information about the inhibited polymerases has been added at the beginning of Results, where we first discuss the mechanism of action of aphidicolin. Our prior work has used more agents than just APH with similar effects on SV induction, so the more general term replication stress is appropriate in Introduction.

4. Please explain what is drawn from Fig. S9 results. Why does ART558 have an effect in TMEJ-deficient cells? Also, recent work suggested the involvement of pol delta in TMEJ. The authors may want to check if aphidicolin affects TMEJ using the Fig. S9 assay method.

We apologize for the lack of clarity, all of Figure S9 was performed with wild-type cells, which we have clarified in text and labeling.

The point about PolD is interesting. To our understanding, its proposed role in TMEJ is mainly as a 3' nuclease, so we wouldn't anticipate an effect of low-dose APH through PolD. Indeed, following on a response to Reviewer 2 in our first revision, we found that SV junctions fully consistent with TMEJ formed in M phase with the same structure and frequency when comparing no APH in M to the highest, fully polymerase-inhibiting dose of APH. Thus, APH did not block M-phase junction formation through inhibition of any target polymerase, whether one used in MiDAS or TMEJ.

5. The Abstract should be toned down as the authors overextend their conclusions. For example, 'as it occurred in cells' should be cautiously be phrased as 'as it occurred in human cell lines'.

The text has been changed as requested. Importantly, the key emphasis point of the sentence is that we could measure SV junction formation “as it occurred” without any cell

outgrowth. The key point is the timing of SV measurements we were able to make, not the specific cells in use.

6. In the first sentence of the third paragraph of the Introduction, the authors state that “... an experimental model for non-recurrent CNV formation”. However, this is not applicable to all non-recurrent CNV sites and therefore needs to be rephrased. Also, in the last sentence (L99), “in normal and cancer cells” should be removed.

The nature of any experimental model is that it is incomplete and provides a specific tractable approach to study a more general problem, with all appropriate caution when generalizing from model to biology. If our work applied to all non-recurrent CNV sites (the entire genome) it wouldn't be a model, so the text is accurate as written.

The other sentence now reads “...provide important insights into possible genome-wide SV mutation mechanisms in normal and cancer cells”, where it is important to highlight both situations where SVs of the general class we study are known to form. The sentence does not claim that we studied normal cells and cancer *per se*, although we did study both normal and cancer cell lines.

7. The authors need to point out the limitations of their study; otherwise, the misleading aspects can have a negative effect in this field. For example, creating so many aphidicolin breaks may not be physiologic. Also, there is no evidence that supports that their findings are true for the entire cell cycle and for normal cells.

We already pointed out limitations of our study throughout Results and Discussion and invited further work to resolve them.

Regarding the non-physiologic nature of aphidicolin, we have modified a sentence in Discussion to discuss not only the size of affected loci but also the intensity (number of loci). The reviewer's assumption about the number of breaks may not be correct – if *de novo* SVs are an indication, we do not anticipate an excessive number of DSBs in any given cell as our data over many publications is most consistent with zero to one SVs forming per cell under the treatment conditions used.

Regarding the cell cycle, we provide extensive data on S, G2 and M phases. As discussed in our previous response, we cannot currently address SV formation in G1 and state this limitation clearly in Discussion and emphasize the need for future work. Any replication-associated SV formation in G1 must occur after a cell would pass from a stressed S-phase through M, and our data clearly show a bolus of SV formation in M-phase. Thus, any G1 effect would be superimposed on a strong M effect.

As noted above, our work is an experimental model, so cannot directly conclude what happens in normal cells in biology, which could not be studied in the same way. That is the nature of experimental inference. We have checked that whenever we extrapolate from experimental results to biology we use appropriate words such as “propose”, “suggest”,

“may”, and “possibility”. There are only a few such locations in the text at the end of the Abstract and the last Introduction and Discussion paragraphs.

Reviewer #3:

The revised manuscript from Wilson and colleagues is improved in clarity, and the points that I asked to be discussed are now included. The main experiment I was asking for was a G1 analysis of SVs. I do understand however the arguments made by the authors that this is a massive undertaking, and somehow out of the scope of the manuscript. The current analysis and findings during mitosis are, per se, sufficient to warrant publication. In conclusion, I believe that the modifications made by the authors are satisfactory and I support the publication of this revised manuscript in Nature Communications.

We thank the reviewer for their endorsement of our manuscript focused on SV formation in M-phase.

REVIEWER 4 COMMENTS

Reviewer 4 comments are in black, our responses are in blue.

In the 3rd revision manuscript:

- first revision changes in the manuscript are marked in red text
- second revision changes are marked in blue text
- third revision changes are marked in purple text

The authors provide a timely, definitive identification of M-phase activity of Pol θ as a major determinant of replication stress-induced SV's at CFS. The following suggested changes could be considered.

1) The authors are appropriately inclusive of “singlicate” data, as these less stringent examples of their data nevertheless help argue for consistency across e.g. cell line models. In some cases (e.g. Fig 1E), this data might be better moved to the supplement.

Many instances of single-replicate data are already in Supplementary, with two exceptions.

Some combinations of manipulations have one replicate of the no-APH control. In all cases, the same cell line has other no-APH replicates in other combinations that always showed the same low baseline level of SV formation. We never observed significant SV formation in any cell line without APH induction over many experiments. In panels with single controls, the critical point is the difference between APH-treated combinations, not APH induction of SVs which is abundantly established. For these reasons, we think the presentation that retains these controls in the main figures is most helpful to readers.

In Figures 1D and 1E, GM12878 cells have a single replicate in the asynchronous (but otherwise unmanipulated) panel. The plot makes clear there is a single data point but emphasizes that we see similar patterns across cell lines. Because Figure 1E sets up significant GM12878 junction data in Figure 1F and is plotted in a manner parallel to Supplementary Figures 1E-G, it would disrupt the presentation to move just those GM12878 bars to Supplementary, so we retained plots as they were. Again, it is readily apparent how many replicates are reported in SV Frequency plots, including in the revised legends, and other cell lines in Figure 1E have more replicates. To draw attention to these issues, we added text to the second Results paragraph that GM12878 has other replicates in later conditions that corroborate deletion SV induction in this cell line.

We removed svCapture statistical comparisons when only one independent biological replicate is available. We retain error bars and significance assessments for two independent biological replicates. Critically, for svCapture SV Frequency panels, each data point itself comprises hundreds to even thousands of SV observations, which is reflected in the negative-binomial-based statistics we used and describe in detail in Methods.

Additionally, per journal format it would be helpful to include in the figure legend a little more of the methods section detail regarding the applied statistical analyses; the current comment “...with selected Poisson-based intergroup p-values...” is a little vague.

We expanded the details of the methods and plotting in Figure 1 to help the reader more quickly understand the statistical data analysis, in addition to various additions requested in the publication checklist. This includes emphasizing that most comparisons were based on a negative binomial, i.e., over-dispersed Poisson, generalized linear regression model.

2) The authors should probably add a comment in the introduction regarding past work arguing the extent low-dose aphidicolin is a generalizable model for induction of replication stress and SV formation at CFS.

We added text to the Introduction to support that CFS instability has most often been studied with low-dose APH and to emphasize the point previously supported only by citations that genomic sequelae induced by APH at CFSs have been observed in other ways, including using other chemical agents, spontaneously, and in human cancers.

Additionally, it may be worth expanding in the discussion as to the importance of DSBs as the likely Pol θ engaging intermediate when considering their model (current mention of MUS81 and GEN1) vs. engagement of Pol θ in repair of gaps using a different source of replication stress (BRCA deficiency; see e.g. Belan et al, Mol. Cell). Additional speculation as to the possibility that different types of replication stress, and the different types of damage they generate (gaps, one ended breaks, and two ended breaks), help determine the repair pathways engaged is perhaps worth mentioning.

We are very interested in the proximate lesions leading to SV formation at CFSs and whether it is the same or different based on the mode of replication stress, although the current manuscript and data focus on downstream events. Given the role of end joining, as evidenced by both repair inhibition and junction structures, it is difficult to project how gaps would lead to SVs without an intervening DSB intermediate, but of course such a DSB could be generated in different ways. We have tersely elaborated on the ways that POLQ might act at different upstream lesions leading to different outcomes, including referencing the work of Belan et al. addressing the role of POLQ at gaps. There is a limit how deeply we can or should engage in such speculation, however, given that our data don't yet address upstream lesion processing leading to SVs.